# Transfer Learning with Kernel Methods

Adityanarayanan Radhakrishnan[1,2,3], Max Ruiz Luyten[1,3], Neha Prasad[1] & Caroline Uhler ®[1,2] ✉

Transfer learning refers to the process of adapting a model trained on a source task to a target task. While kernel methods are conceptually and computationally simple models that are competitive on a variety of tasks, it has been unclear how to develop scalable kernel-based transfer learning methods across general source and target tasks with possibly differing label dimensions. In this work, we propose a transfer learning framework for kernel methods by projecting and translating the source model to the target task. We demonstrate the effectiveness of our framework in applications to image classification and virtual drug screening. For both applications, we identify simple scaling laws that characterize the performance of transfer-learned kernels as a function of the number of target examples. We explain this phenomenon in a simplified linear setting, where we are able to derive the exact scaling laws.

Transfer learning refers to the machine learning problem of utilizing knowledge from a source task to improve performance on a target task. Recent approaches to transfer learning have achieved tremendous empirical success in many applications, including in computer vision[1,2], natural language processing[3–5], and the biomedical field[6,7]. Since transfer learning approaches generally rely on complex deep neural networks, it can be difficult to characterize when and why they work[8]. Kernel methods[9] are conceptually and computationally simple machine learning models that have been found to be competitive with neural networks on a variety of tasks, including image classification[10–12] and drug screening[12]. Their simplicity stems from the fact that training a kernel method involves performing linear regression after transforming the data. There has been renewed interest in kernels due to a recently established equivalence between wide neural networks and kernel methods[13,14], which has led to the development of modern, neural tangent kernels (NTKs) that are competitive with neural networks. Given their simplicity and effectiveness, kernel methods could provide a powerful approach for transfer learning and also help characterize when transfer learning between a source and target task would be beneficial.

Yet, developing scalable algorithms for transfer learning with kernel methods for general source and target tasks with possibly differing label dimensions has been an open problem. In particular, while there is a standard transfer learning approach for neural networks that involves replacing and re-training the last layer of a pre-trained network, there is no known corresponding operation for kernels. Prior works on transfer learning with kernels focus on applications in which the source and target tasks have the same label sets[15–20]. Examples include predicting stock returns for a given sector based on returns available for other sectors[16] or predicting electricity consumption for certain zones of the United States based on the consumption in other zones[17]. These methods are not applicable to general source and target tasks with differing label dimensions, including classical transfer learning applications such as using a model trained to classify between thousands of objects to subsequently classify new objects. There are also various works on using kernels for multi-task learning problems[21–23], which, in the context of transfer learning, assume that source and target data are available at the time of training the source model. These methods can be computationally expensive since they involve computing matrix-valued kernels, where the number of rows/columns is equal to the number of labels. As a consequence, for a kernel trained on ImageNet32[24] with 1000 possible labels, a matrix-valued kernel would involve $10^6$ times more compute than a classical kernel method. Prior works also develop kernel-based methods for learning a re-weighting or transformation that captures similarities across source and target data distributions[25–27]. Such a transformation is typically learned by solving an optimization problem that involves materializing the full training kernel matrix, which can be computationally prohibitive (e.g., for a dataset with a million samples, this would require more than 3.5 terabytes of memory).

In this work, we present a general, scalable framework for performing transfer learning with kernel methods. Unlike prior work, our

[1]Massachusetts Institute of Technology, Cambridge, MA, USA. [2]Broad Institute of MIT and Harvard, Cambridge, MA, USA. [3]These authors contributed equally: Adityanarayanan Radhakrishnan, Max Ruiz Luyten. ✉e-mail: cuhler@mit.edu

framework enables transfer learning for kernels regardless of whether the source and target tasks have the same or differing label sets. Furthermore, like for transfer learning methodology for neural networks, our framework allows transferring to a variety of target tasks after training a kernel method only once on a source task.

The key components of our transfer learning framework are: Train a kernel method on a source dataset and then apply the following operations to transfer the model to the target task.

- Projection. We apply the trained source kernel to each sample in the target dataset and then train a secondary model on these source predictions to solve the target task; see Fig. 1a.
- Translation. When the source and target tasks have the same label sets, we train a correction term that is added to the source model to adapt it to the target task; see Fig. 1c.

We note that while these two operations are general and can be applied to any predictor, we focus on using them in conjunction with kernel methods due to their conceptual simplicity, effectiveness, and flexibility, in particular given that they include infinite-width neural networks[13] as a subclass. Moreover, the closed-form solutions provided by kernel methods also enable a theoretical analysis of transfer learning. Projection and translation are motivated by operations that are standardly used for transfer learning using neural networks. Namely, projection corresponds to adding layers at the end of a neural network trained on a source task and then training the weights in these new layers on the target task. And when approximating a neural network by its linearization around the initial parameters[13,28], transfer learning by tuning the weights of the source model on the target task is equivalent to performing translation. Our formulation of projection and translation makes these operations compatible with recent preconditioned gradient descent kernel regression solvers such as

EigenPro[29], thereby allowing our framework to easily scale to datasets such as ImageNet32 with over one million samples.

Projection is effective when the source model predictions contain information regarding the target labels. We will demonstrate that this is the case in image classification tasks in which the predictions of a classifier trained to distinguish between a thousand objects in ImageNet32[24] provides information regarding the labels of images in other datasets, such as street view house numbers (SVHN)[30]; see Fig. 1b. In particular, we will show across 23 different source and target task combinations that kernels transferred using our approach achieve up to a 10% increase in accuracy over kernels trained on target tasks directly.

On the other hand, translation is effective when the predictions of the source model can be corrected to match the labels of the target task via an additive term. We will show that this is the case in virtual drug screening in which a model trained to predict the effect of a drug on one cell line can be adjusted to capture the effect on a new cell line; see Fig. 1d. In particular, we will show that our transfer learning approach provides an improvement to prior kernel method predictors[12] even when transferring to cell lines and drugs not present in the source task.

Interestingly, we observe that for both applications, image classification and virtual drug screening, transfer learned kernel methods follow simple scaling laws; i.e., how the number of available target samples effects the performance on the target task can be accurately modelled. As a consequence, our work provides a simple method for estimating the impact of collecting more target samples on the performance of the transfer learned kernel predictors. In the simplified setting of transfer learning with linear kernel methods we are able to mathematically derive the scaling laws, thereby providing a mathematical basis for the empirical observations. To do so, we obtain exact

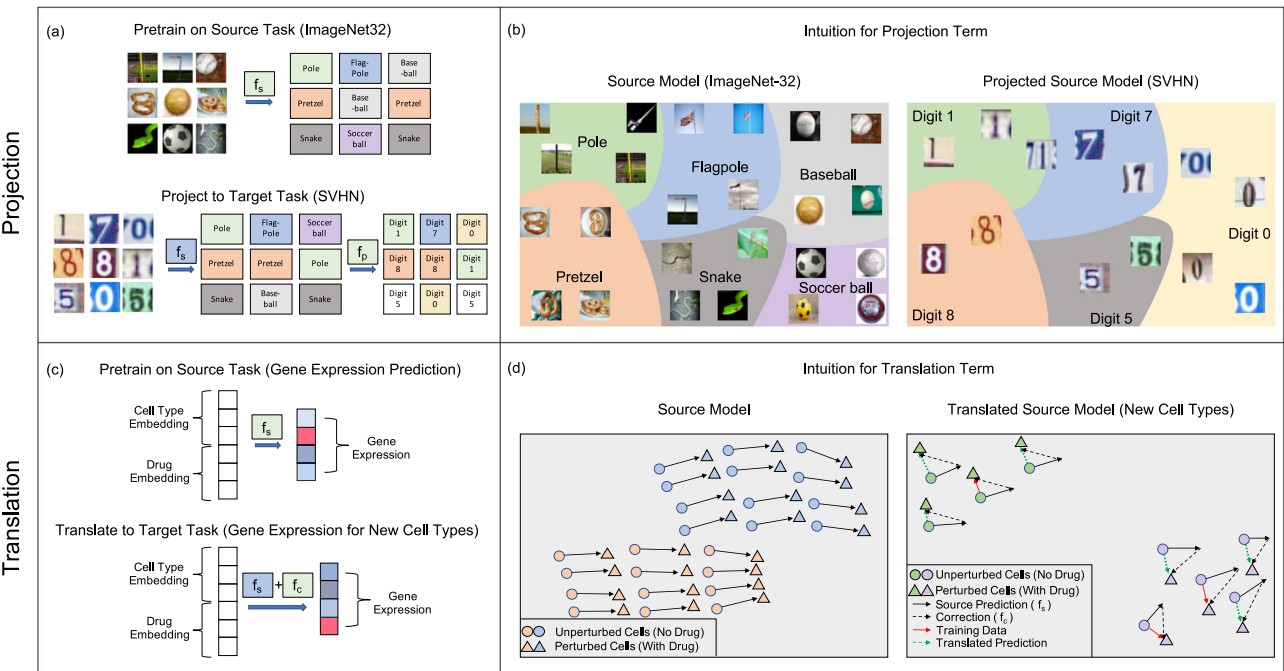

**Fig. 1 | Our framework for transfer learning with kernel methods for supervised learning tasks.** After training a kernel method on a source task, we transfer the source model to the target task via a combination of projection and translation operations. **a** Projection involves training a second kernel method on the predictions of the source model on the target data, as is shown for image classification between natural images and house numbers. **b** Projection is effective when the predictions of the source model on target examples provide useful information about target labels; e.g., a model trained to classify natural images may be able to distinguish the images of zeros from ones by using the similarity of zeros to balls and ones to poles. **c** Translation involves adding a correction term to the source model, as is shown for predicting the effect of a drug on a cell line. **d** Translation is effective when the predictions of the source model can be additively corrected to match labels in the target data; e.g., the predictions of a model trained to predict the effect of drugs on one cell line may be additively adjustable to predict the effect on new cell lines.

 

non-asymptotic formulas for the risk of the projected and translated predictors. Our non-asymptotic analysis is in contrast to a large number of prior works analyzing multitask learning algorithms[31–34] and meta-learning algorithms[35,36], which provide generalization bounds establishing statistical consistency of these methods but do not provide an explicit form of the risk, which is required for deriving explicit scaling laws. Overall, our work demonstrates that transfer learning with kernel methods between general source and target tasks is possible and demonstrates the simplicity and effectiveness of the proposed method on a variety of important applications.

## Results

In the following, we present our framework for transfer learning with kernel methods more formally. Since kernel methods are fundamental to this work, we start with a brief review.

Given training examples $X = [x^{(1)}, \ldots, x^{(n)}] \in \mathbb{R}^{d \times n}$, corresponding labels $y = [y^{(1)}, \ldots, y^{(n)}] \in \mathbb{R}^{1 \times n}$, a standard nonlinear approach to fitting the training data is to train a kernel machine[9]. This approach involves first transforming the data, $\{x^{(i)}\}_{i=1}^{n}$, with a feature map, $\psi$, and then performing linear regression. To avoid defining and working with feature maps explicitly, kernel machines rely on a kernel function, $K : \mathbb{R}^{d} \times \mathbb{R}^{d} \to \mathbb{R}$, which corresponds to taking inner products of the transformed data, i.e., $K(x^{(i)}, x^{(j)}) = \langle \psi(x^{(i)}), \psi(x^{(j)}) \rangle$. The trained kernel machine predictor uses the kernel instead of the feature map and is given by:

$$\hat{f}(x) = \alpha K(X, x), \text{ where } \alpha = \underset{w \in \mathbb{R}^{1 \times n}}{\operatorname{argmin}} \|y - w K_n\|_2^2, \quad (1)$$

and $K_n \in \mathbb{R}^{n \times n}$ with $(K_n)_{i,j} = K(x^{(i)}, x^{(j)})$ and $K(X, x) \in \mathbb{R}^{n}$ with $K(X, x)_i = K(x^{(i)}, x)$. Note that for datasets with over $10^5$ samples, computing the exact minimizer $\alpha$ is computationally prohibitive, and we instead use fast, approximate iterative solvers such as EigenPro[29]. For a more detailed description of kernel methods see SI Note 1.

For the experiments in this work, we utilize a variety of kernel functions. In particular, we consider the classical Laplace kernel given by $K(x, \tilde{x}) = \exp(-L \| x - \tilde{x} \|_2)$, which is a standard benchmark kernel that has been widely used for image classification and speech recognition[29]. In addition, we consider recently discovered kernels that correspond to infinitely wide neural networks. While there is an emerging understanding that increasingly wider neural networks generalize better[37,38], such models are generally computationally difficult to train. Remarkably, recent work identified conditions under which neural networks in the limit of infinite width implement kernel machines; the corresponding kernel is known as the Neural Tangent Kernel (NTK)[13]. In the following, we use the NTK corresponding to training an infinitely wide ReLU fully connected network[13] and also the convolutional NTK (CNTK) corresponding to training an infinitely wide ReLU convolutional network[14]. We chose to use the CNTK without global average pooling (GAP)[14] for our experiments. While the CNTK model with GAP as well as the models considered in[39] give higher accuracy on image datasets, they are computationally prohibitive to compute for our large-scale experiments. For example, a CNTK with GAP is estimated to take 1200 GPU hours for 50k training samples[11].

Unlike the usual supervised learning setting where we train a predictor on a single domain, we will consider the following transfer learning setting from[40], which involves two domains: (1) a source with domain $\mathcal{X}_s$ and data distribution $\mathbb{P}_s$; and (2) a target with domain $\mathcal{X}_t$ and data distribution $\mathbb{P}_t$. The goal is to learn a model for a target task $f_t : \mathcal{X}_t \to \mathcal{Y}_t$ by making use of a model trained on a source task $f_s : \mathcal{X}_s \to \mathcal{Y}_s$. We let $c_s$ and $c_t$ denote the dimensionality of $\mathcal{Y}_s$ and $\mathcal{Y}_t$ respectively, i.e. for image classification these denote the number of classes in the source and target. Lastly, we let $(X_s, y_s) \in \mathcal{X}_s^{n_s} \times \mathcal{Y}_s^{n_s}$ and $(X_t, y_t) \in \mathcal{X}_t^{n_t} \times \mathcal{Y}_t^{n_t}$ denote the source and target dataset, respectively.

Throughout this work, we assume that the source and target domains are equal ($\mathcal{X}_s = \mathcal{X}_t$), but that the data distributions differ ($\mathbb{P}_s \neq \mathbb{P}_t$).

Our work is concerned with the recovery of $f_t$ by transferring a model, $\hat{f}_s$, that is learned by training a kernel machine on the source dataset. To enable transfer learning with kernels, we propose the use of two methods, projection and translation. We first describe these methods individually and demonstrate their performance on transfer learning for image classification using kernel methods. For each method, we empirically establish scaling laws relating the quantities $n_s, n_t, c_s, c_t$ to the performance boost given by transfer learning, and we also derive explicit scaling laws when $f_t, f_s$ are linear maps. We then utilize a combination of the two methods to perform transfer learning in an application to virtual drug screening.

### Transfer learning via projection

Projection involves learning a map from source model predictions to target labels and is thus particularly suited for situations where the number of labels in the source task $c_s$ is much larger than the number of labels in the target task $c_t$.

**Definition 1.** Given a source dataset $(X_s, y_s)$ and a target dataset $(X_t, y_t)$, the projected predictor, $\hat{f}_t$, is given by:

$$\hat{f}_t(x) = \hat{f}_p(\hat{f}_s(x)), \text{ where } \hat{f}_p := \underset{\{f : \mathcal{Y}_s \to \mathcal{Y}_t\}}{\operatorname{argmin}} \|y_t - f(\hat{f}_s(X_t))\|^2, \quad (2)$$

where $\hat{f}_s$ is a predictor trained on the source dataset. When there are infinitely many possible values for the parameterized function $\hat{f}_p$, we consider the minimum norm solution.

While Definition 1 is applicable to any machine learning method, we focus on predictors $\hat{f}_s$ and $\hat{f}_p$ parameterized by kernel machines given their conceptual and computational simplicity. As illustrated in Fig. 1a and b, projection is effective when the predictions of the source model already provide useful information for the target task.

Kernel-based image classifier performance improves with projection. We now demonstrate the effectiveness of projected kernel predictors for image classification. In particular, we first train kernels to classify among 1000 objects across 1.28 million images in ImageNet32 and then transfer these models to 4 different target image classification datasets: CIFAR10[41], Oxford 102 Flowers[42], Describable Textures Datasets[43], and SVHN[30]. We selected these datasets since they cover a variety of transfer learning settings, i.e. all of the CIFAR10 classes are in ImageNet32, ImageNet32 contains only 2 flower classes, and none of DTD and SVHN classes are in ImageNet32. A full description of the datasets is provided in Methods.

For all datasets, we compare the performance of 3 kernels (the Laplace kernel, NTK, and CNTK) when trained just on the target task, i.e. the *baseline predictor*, and when transferred via projection from ImageNet32. Training details for all kernels are provided in Methods. In Fig. 2a, we showcase the improvement of projected kernel predictors over baseline predictors across all datasets and kernels. We observe that projection yields a sizeable increase in accuracy (up to 10%) on the target tasks, thereby highlighting the effectiveness of this method. It is remarkable that this performance increase is observed even for transferring to Oxford 102 Flowers or DTD, datasets that have little to no overlap with images in ImageNet32.

In SI Fig. S1a, we compare our results with those of a finite-width neural network analog of the (infinite-width) CNTK where all layers of the source network are fine-tuned on the target task using the standard cross-entropy loss[44] and the Adam optimizer[45]. We observe that the performance gap between transfer-learned finite-width neural networks and the projected CNTK is largely influenced by the performance gap between these models on ImageNet32. In fact, in SI Fig. S1a, we show that finite-width neural networks trained to the same test accuracy on ImageNet32 as the (infinite-width) CNTK yield lower

 

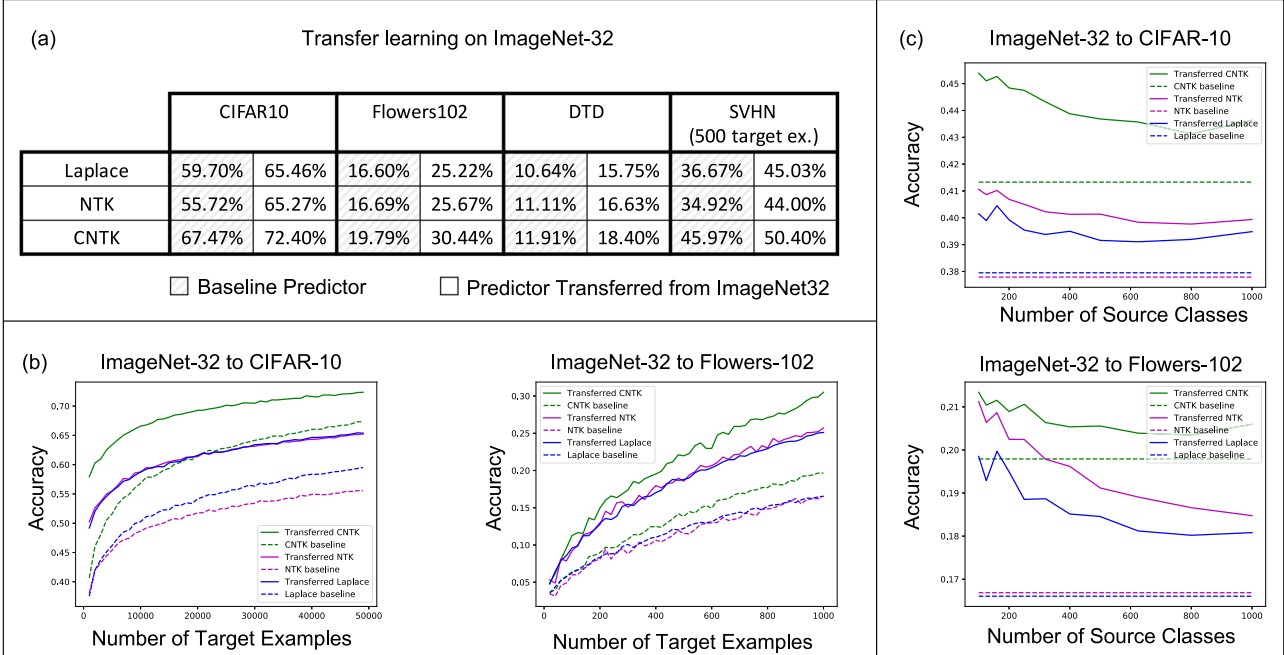

**Fig. 2 | Analysis of transfer learning with kernels trained on ImageNet32 to CIFAR10, Oxford 102 Flowers, DTD, and a subset of SVHN.** All curves in (**b**, **c**) are averaged over 3 random seeds. **a** Comparison of the transferred kernel predictor test accuracy (green) to the test accuracy of the baseline kernel predictors trained directly on the target tasks (red). In all cases, the transferred kernel predictors outperform the baseline predictors and the difference in performance is as high as 10%. **b** Test accuracy of the transferred and baseline predictors as a function of the number of target examples. These curves, which quantitatively describe the benefit of collecting more target examples, follow simple logarithmic trends ($R^2 > .95$). **c** Performance of the transferred kernel methods decreases when increasing the number of source classes but keeping the total number of source examples fixed. Corresponding plots for DTD and SVHN are in SI Fig. S2.

performance than the CNTK when transferred to target image classification tasks.

The computational simplicity of kernel methods allows us to compute scaling laws for the projected predictors. In Fig. 2b, we analyze how the performance of projected kernel methods varies as a function of the number of target examples, $n_t$, for CIFAR10 and Oxford 102 Flowers. The results for DTD and SVHN are presented in SI Fig. S2a and b. For all target datasets, we observe that the accuracy of the projected predictors follows a simple logarithmic trend given by the curve $a \log n_t + b$ for constants $a, b$ ($R^2$ values on all datasets are above 0.95). By fitting this curve on the accuracy corresponding to just the smallest five values of $n_t$, we are able to predict the accuracy of the projected predictors within 2% of the reported accuracy for large values of $n_t$ (see Methods, SI Fig. S4). The robustness of this fit across many target tasks illustrates the practicality of the transferred kernel methods for estimating the number of target examples needed to achieve a given accuracy. Additional results on the scaling laws upon varying the number of source examples per class are presented in SI Fig. S3 for transferring between ImageNet32 and CIFAR10. In general, we observe that the performance increases as the number of source training examples per class increases, which is expected given the similarity of source and target tasks.

Lastly, we analyze the impact of increasing the number of classes while keeping the total number of source training examples fixed at 40k. Figure 2c shows that having few samples for each class can be worse than having a few classes with many samples. This may be expected for datasets such as CIFAR10, where the classes overlap with the ImageNet32 classes: having few classes with more examples that overlap with CIFAR10 should be better than having many classes with fewer examples per class and less overlap with CIFAR10. A similar trend can be observed for DTD, but interestingly, the trend differs for SVHN, indicating that SVHN images can be better classified by projecting from a variety of ImageNet32 classes (see SI Fig. S2).

## Transfer learning via translation

While projection involves composing a map with the source model, the second component of our framework, translation, involves adding a map to the source model as follows.

**Definition 2.** Given a source dataset $(X_s, y_s)$ and a target dataset $(X_t, y_t)$, the translated predictor, $\hat{f}_t$, is given by:

$$\hat{f}_t(x) = \hat{f}_s(x) + \hat{f}_c(x), \text{ where } \hat{f}_c = \underset{\{f:\mathcal{X}_t \to \mathcal{Y}_t\}}{\operatorname{argmin}} \|y_t - \hat{f}_s(X_t) - f(X_t)\|^2, \quad (3)$$

where $\hat{f}_s$ is a predictor trained on the source dataset. When there are infinitely many possible values for the parameterized function $\hat{f}_c$, we consider the minimum norm solution.

Translated predictors correspond to first utilizing the trained source model directly on the target task and then applying a correction, $\hat{f}_c$, which is learned by training a model on the corrected labels, $y_t - \hat{f}_s(X_t)$. Like for the projected predictors, translated predictors can be implemented using any machine learning model, including kernel methods. When the predictors $\hat{f}_s$ and $\hat{f}_c$ are parameterized by linear models, translated predictors correspond to training a target predictor with weights initialized by those of the trained source predictor (proof in SI Note 4). We note that training translated predictors is also a new form of boosting[46] between the source and target dataset, since the correction term accounts for the error of the source model on the target task. Lastly, we note that while the formulation given in Definition 2 requires the source and target tasks to have the same label dimension, projection and translation can be naturally combined to overcome this restriction.

Kernel-based image classifier performance improves with translation. We now demonstrate that the translated predictors are particularly well-suited for correcting kernel methods to handle distribution shifts in images. Namely, we consider the task of transferring a source model trained on CIFAR10 to corrupted CIFAR10

images in CIFAR10-C[47]. CIFAR10-C consists of the test images in CIFAR10, but the images are corrupted by one of 19 different perturbations, such as adjusting image contrast and introducing natural artifacts such as snow or frost. In our experiments, we select the 10k images of CIFAR10-C with the highest level of perturbation, and we reserve 9k images of each perturbation for training and 1k images for testing. In SI Fig. S5, we additionally analyze translating kernels from subsets of ImageNet32 to CIFAR10.

Again, we compare the performance of the three kernel methods considered for projection, but along with the accuracy of the translated predictor and baseline predictor, we also report the accuracy of the *source predictor*, which is given by using the source model directly on the target task. In Fig. 3a and SI Fig. S6, we show that the translated predictors outperform the baseline and source predictors on all 19 perturbations. Interestingly, even for corruptions such as contrast and fog where the source predictor is worse than the baseline predictor, the translated predictor outperforms all other kernel predictors by up to 11%. In SI Fig. S6, we show that for these corruptions, the translated kernel predictors also outperform the projected kernel predictors trained on CIFAR10. In SI Fig. S1b, we additionally compare with the performance of a finite-width analog of the CNTK by fine-tuning all layers on the target task with cross-entropy loss and the Adam optimizer. We observe that the translated kernel methods outperform the corresponding neural networks. Remarkably kernels translated from CIFAR10 can even outperform fine-tuning a neural network pre-trained on ImageNet32 for several perturbations (see SI Fig. S1c). In SI Fig. S7, we additionally demonstrate the effectiveness of our translation methodology over prior transfer learning methods using multiple kernel learning (see Methods for further details).

Analogously to our analysis of the projected predictors, we visualize how the accuracy of the translated predictors is affected by the number of target examples, $n_t$, for a subset of corruptions shown in Fig. 3b. We observe that the performance of the translated predictors is heavily influenced by the performance of the source predictor. For example, as shown in Fig. 3b for the brightness perturbation, where the

source predictor already achieves an accuracy of 60.80%, the translated predictors achieve an accuracy of above 60% when trained on only 10 target training samples. For the examples of the contrast and fog corruptions, Fig. 3b also shows that very few target examples allow the translated predictors to outperform the source predictors (e.g., by up to 5% for only 200 target examples). Overall, our results showcase that translation is effective at adapting kernel methods to distribution shifts in image classification.

### Transfer learning via projection and translation in virtual drug screening

We now demonstrate the effectiveness of projection and translation for the use of kernel methods for virtual drug screening. A common problem in drug screening is that experimentally measuring many different drug and cell line combinations is both costly and time-consuming. The goal of virtual drug screening approaches is to computationally identify promising candidates for experimental validation. Such approaches involve training models on existing experimental data to then impute the effect of drugs on cell lines for which there was no experimental data.

The CMAP dataset[48] is a large-scale, publicly available drug screen containing measurements of 978 landmark genes for 116,228 combinations of 20,336 drugs (molecular compounds) and 70 cell lines. This dataset has been an important resource for drug screening[49,50]. CMAP also contains data on genetic perturbations; but in this work, we focus on imputing the effect of chemical perturbations only. Prior work for virtual drug screening demonstrated the effectiveness of low-rank tensor completion and nearest neighbor predictors for imputing the effect of unseen drug and cell line combinations in CMAP[51]. However, these methods crucially rely on the assumption that for each drug there is at least one measurement for every cell line, which is not the case when considering new chemical compounds. To overcome this issue, recent work[12] introduced kernel methods for drug screening using the NTK to predict gene expression vectors from drug and cell line embeddings, which capture the similarity between drugs and cell lines.

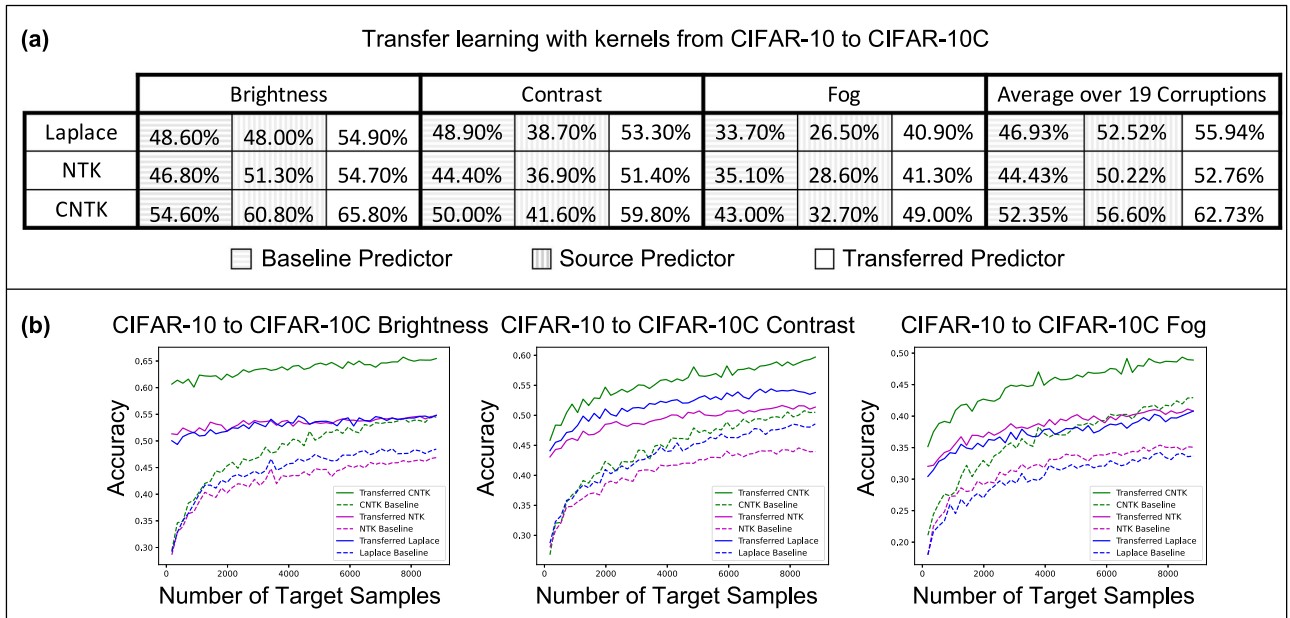

**Fig. 3 | Transferring kernel methods from CIFAR10 to adapt to 19 different corruptions in CIFAR10-C. a** Test accuracy of baseline kernel method (red), using source predictor given by directly applying the kernel trained on CIFAR10 to CIFAR10-C (gray), and transferred kernel method (green). The transferred kernel method outperforms the other models on all 19 corruptions and even improves on

the baseline kernel method when the source predictor exhibits a decrease in performance. Additional results are presented in SI Fig. S6. **b** Performance of the transferred and baseline kernel predictors as a function of the number of target examples. The transferred kernel method can outperform both source and baseline predictors even when transferred using as little as 200 target examples.

In the following, we demonstrate that the NTK predictor can be transferred to improve gene expression imputation for drug and cell line combinations, even in cases where neither the particular drug nor the particular cell line were available when training the source model. To utilize the framework of[12], we use the control gene expression vector as cell line embedding and the 1024-bit circular fingerprints from[52] as drug embedding. All pre-processing of the CMAP gene expression vectors is described in Methods. For the source task, we train the NTK to predict gene expression for the 54,444 drug and cell line combinations corresponding to the 65 cell lines with the least drug availability in CMAP. We then impute the gene expression for each of the 5 cell lines (A375, A549, MCF7, PC3, VCAP) with the most drug availability. We chose these data splits in order to have sufficient target samples to analyze model performance as a function of the number of target samples. In our analysis of the transferred NTK, we always consider transfer to a new cell line, and we stratify by whether a drug in the target task was already available in the source task. For this application we combine projection and translation into one predictor as follows.

**Definition 3.** Given a source dataset $(X_s, y_s)$ and a target dataset $(X_t, y_t)$, the projected and translated predictor, $\hat{f}_{pt}$, is given by:

$$\hat{f}_{pt}(x) = \hat{f}\left(\left[\hat{f}_s(x) \mid x\right]\right), \text{where } \hat{f} = \underset{f:\mathcal{Y}_s \times \mathcal{X}_t \to \mathcal{Y}_s}{\text{argmin}} \left\| y_t - f\left(\left[\hat{f}_s(X_t) \mid X_t\right]\right) \right\|^2,$$
(4)

where $\hat{f}_s$ is a predictor trained on the source dataset and $\left[\hat{f}_s(x) \mid x\right] \in \mathcal{Y}_s \times \mathcal{X}_t$ is the concatenation of $\hat{f}_s(x)$ and $x$.

Note that if we omit $x, X_t$ in the concatenation above, we get the projected predictor, and if $f$ is additive in its arguments, i.e., if $f([\hat{f}_s(x) \mid x]) = \hat{f}_s(x) + x$, we get the translated predictor. Generally, $\hat{f}_s(x)$ and $x$ can correspond to different modalities (e.g., class label vectors and images), but in the case of drug screening, both correspond to gene expression vectors of the same dimension. Thus, combining projection and translation is natural in this context.

Figure 4 a and b show that the transferred kernel predictors outperform both, the baseline model from[12] as well as imputation by mean (over each cell line) gene expression across three different metrics ($R^2$, cosine similarity, and Pearson r value) on both tasks (i.e., transferring to drugs that were seen in the source task as well as completely new drugs). All metrics and training details are presented in Methods. Interestingly, the transferred kernel methods provide a boost over the baseline kernel methods even when transferring to new cell lines and new drugs. But as expected, we note that the increase in performance is greater when transferring to drug and cell line combinations for which the drug was available in the source task. Figure 4c and d show that the transferred kernels again follow simple logarithmic scaling laws (fitting a logarithmic model to the red and green curves yields $R^2 > 0.9$). We note that the transferred NTKs have better-scaling coefficients than the baseline models, thereby implying that the performance gap between the transferred NTK and the baseline NTK grows as more target examples are collected until the performance of the transferred NTK saturates at its maximum possible value. In Fig. 4e and f, we visualize the performance of the transferred NTK in relation to the top 2 principal components of gene expression for drug and cell line combinations. We generally observe that the performance of the NTK is lower for cell and drug combinations that are further from the control, i.e., the unperturbed state. Plots for the other 3 cell lines are presented in SI Fig. S8. In Methods and SI Fig. S9, we show that this approach can also be used for other transfer learning tasks related to virtual drug screening. In particular, we show that the imputed gene expression vectors can be transferred to predict the viability of a drug and cell line combination in the large-scale, publicly available Cancer Dependency Map (DepMap) dataset[53].

## Theoretical analysis of projection and translation in the linear setting

In the following, we provide explicit scaling laws for the performance of projected and translated kernel methods in the linear setting, thereby providing a mathematical basis for the empirical observations in the previous sections.

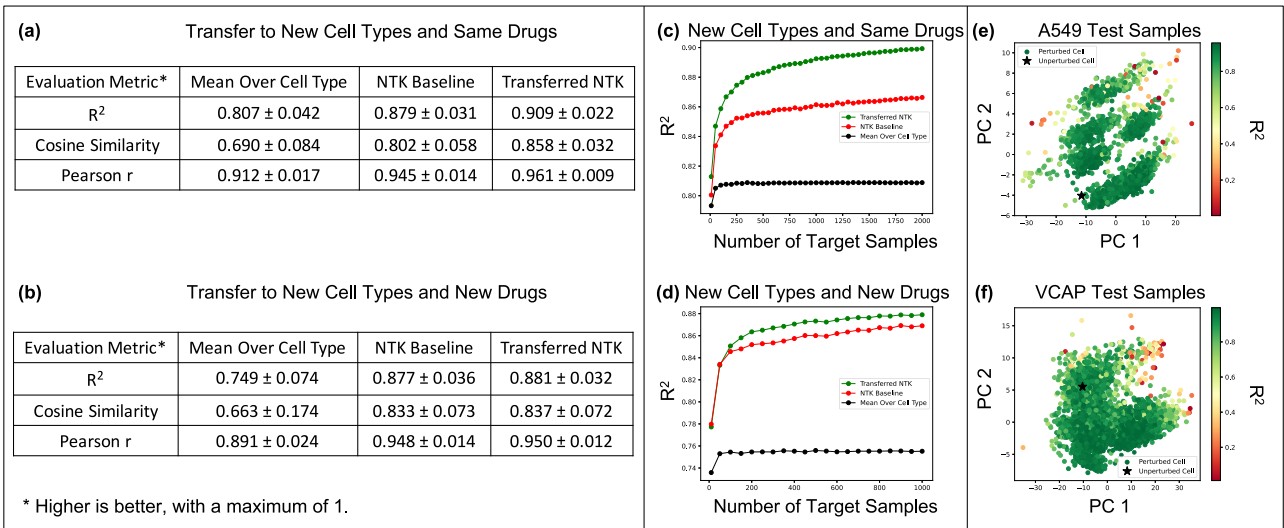

**(a)** Transfer to New Cell Types and Same Drugs

| Evaluation Metric* | Mean Over Cell Type | NTK Baseline | Transferred NTK |
|---|---|---|---|
| $R^2$ | $0.807 \pm 0.042$ | $0.879 \pm 0.031$ | $0.909 \pm 0.022$ |
| Cosine Similarity | $0.690 \pm 0.084$ | $0.802 \pm 0.058$ | $0.858 \pm 0.032$ |
| Pearson r | $0.912 \pm 0.017$ | $0.945 \pm 0.014$ | $0.961 \pm 0.009$ |

**(b)** Transfer to New Cell Types and New Drugs

| Evaluation Metric* | Mean Over Cell Type | NTK Baseline | Transferred NTK |
|---|---|---|---|
| $R^2$ | $0.749 \pm 0.074$ | $0.877 \pm 0.036$ | $0.881 \pm 0.032$ |
| Cosine Similarity | $0.663 \pm 0.174$ | $0.833 \pm 0.073$ | $0.837 \pm 0.072$ |
| Pearson r | $0.891 \pm 0.024$ | $0.948 \pm 0.014$ | $0.950 \pm 0.012$ |

* Higher is better, with a maximum of 1.

**Fig. 4 | Transferring the NTK trained to predict gene expression for given drug and cell line combinations in CMAP to new drug and cell line combinations. a, b** The transfer learned NTK (green) outperforms imputation by mean over cell line (gray) and previous NTK baseline predictors from[12] across $R^2$, cosine similarity, and Pearson r metrics. All results are averaged over the performance on 5 cell lines and are stratified by whether or not the target data contains drugs that are present in the source data. Error bars indicate standard deviation. **c, d** The transferred kernel method performance follows a logarithmic trend ($R^2 > .9$) as a function of the number of target examples and exhibits a better scaling coefficient than the baselines. The results are averaged over 5 cell lines. **e, f** Visualization of the performance of the transferred NTK in relation to the top two principal components (denoted PC1 and PC2) of gene expression for target drug and cell line combinations. The performance of the NTK is generally lower for cell and drug combinations that are further from the control gene expression for a given cell line. Visualizations for the remaining 3 cell lines are presented in SI Fig. S8.

We derive scaling laws for projected predictors in the following linear setting. We assume that $\mathcal{X} = \mathbb{R}^d$, $\mathcal{Y}_s = \mathbb{R}^{c_s}$, $\mathcal{Y}_t = \mathbb{R}^{c_t}$ and that $f_s$ and $f_t$ are linear maps, i.e., $f_s = \omega_s \in \mathbb{R}^{c_s \times d}$ and $f_t = \omega_t \in \mathbb{R}^{c_t \times d}$. The following results provide a theoretical foundation for the empirical observations regarding the role of the number of source classes and the number of source samples for transfer learning shown in Fig. 2 as well as in[54]. In particular, we will derive scaling laws for the *risk*, or expected test error, of the projected predictor as a function of the number of source examples, $n_s$, target examples, $n_t$, and number of source classes, $c_s$. We note that the risk of a predictor is a standard object of study for understanding generalization in statistical learning theory[55] and defined as follows.

**Definition 4.** Let $\mathbb{P}$ be a probability density on $\mathbb{R}^d$ and let $x, x^{(i)} \sim i.i.d. \mathbb{P}$ for $i = 1, 2, \dots n$. Let $X = [x^{(1)}, \dots, x^{(n)}] \in \mathbb{R}^{d \times n}$ and $y = [w^* x^{(1)}, \dots w^* x^{(n)}] \in \mathbb{R}^{c \times n}$ for $w^* \in \mathbb{R}^{c \times d}$. The risk of a predictor $\hat{w}$ trained on the samples $(X, y)$ is given by

$$\mathcal{R}(\hat{w}) = \mathbb{E}_{x, X}[\|w^* x - \hat{w} x\|_F^2]. \tag{5}$$

By understanding how the risk scales with the number of source examples, target examples, and source classes, we can characterize the settings in which transfer learning is beneficial. As is standard in analyses of the risk of over-parameterized linear regression[56–59], we consider the risk of the minimum norm solution given by

$$\hat{w} = \underset{w}{\mathrm{argmin}} \|y - wX\|_F^2, \text{i.e., } \hat{w} = yX^\dagger, \tag{6}$$

where $X^\dagger$ is the Moore-Penrose inverse of $X$. Theorem 1 establishes a closed form for the risk of the projected predictor $\hat{\omega}_p \hat{\omega}_s$, thereby giving a closed form for the scaling law for transfer learning in the linear setting; the proof is given in SI Note 2.

**Theorem 1.** Let $\mathcal{X} = \mathbb{R}^d$, $\mathcal{Y}_s = \mathbb{R}^{c_s}$, $\mathcal{Y}_t = \mathbb{R}^{c_t}$, and let $\hat{\omega}_s = y_s X_s^\dagger$ and $\hat{\omega}_p = y_t (\hat{\omega}_s X_t)^\dagger$. Assuming that $\mathbb{P}_s$ and $\mathbb{P}_t$ are independent, isotropic distributions on $\mathbb{R}^d$, then the risk $\mathcal{R}(\hat{\omega}_p \hat{\omega}_s)$ is given by

$$\mathcal{R}(\hat{\omega}_p \hat{\omega}_s) = \left[ (C_1 + C_2 K_1) \left(1 - \frac{n_t}{d}\right) + (1 - C_1 - C_2) \right] \|\omega_t\|_F^2 + C_2 K_2 \varepsilon, \tag{7}$$

where $\varepsilon = \|\omega_t (I_{d \times d} - \omega_s^\dagger \omega_s)\|_F^2$ and

$$C_1 = \frac{n_s c_s (d - n_s)}{d(d-1)(d+2)}, \quad C_2 = \frac{n_s[d(n_s+1) - 2]}{d(d-1)(d+2)},$$
$$K_1 = 1 - \frac{n_t(d - c_s)}{(d-1)(d+2)}, \quad K_2 = \frac{n_t}{d} + \frac{n_t(d - n_t)}{(d-1)(d+2)}.$$

The $\varepsilon$ term in Theorem 1 quantifies the similarity between the source and target tasks. For example, if there exists a linear map $\omega_p$ such that $\omega_p \omega_s = \omega_t$, then $\varepsilon = 0$. In the context of classification, this can occur if the target classes are a strict subset of the source classes. Since transfer learning is typically performed between source and target tasks that are similar, we expect $\varepsilon$ to be small. To gain more insights into the behavior of transfer learning using the projected predictor, the following corollary considers the setting where $d \to \infty$ in Theorem 1; the proof is given in SI Note 3.

**Corollary 1.** Let $S = \frac{n_s}{d}$, $T = \frac{n_t}{d}$, $C = \frac{c_s}{d}$ and assume $\|\omega_t\|_F = \Theta(1)$. Under the setting of Theorem 1, if $S, T, C < \infty$ as $d \to \infty$, then:
a. $\mathcal{R}(\hat{\omega}_p \hat{\omega}_s)$ is monotonically decreasing for $S \in [0, 1]$ if $\varepsilon < (1 - C)\|\omega_t\|_F$.
b. If $2S - 1 - ST < 0$, then $\mathcal{R}(\hat{\omega}_p \hat{\omega}_s)$ decreases as $C$ increases.
c. If $S = 1$, then $\mathcal{R}(\hat{\omega}_p \hat{\omega}_s) = (1 - T + TC)\mathcal{R}(\hat{\omega}_t) + \varepsilon T(2 - T)$.
d. If $S = 1$ and $T, C = \Theta(\delta)$, then $\mathcal{R}(\hat{\omega}_p \hat{\omega}_s) = (1 - 2T)\|\omega_t\|_F^2 + 2T\varepsilon + \Theta(\delta^2)$.

Corollary 1 not only formalizes several intuitions regarding transfer learning, but also theoretically corroborates surprising dependencies on the number of source examples, target examples, and source classes that were empirically observed in Fig. 2 for kernels and in[54] for convolutional networks. First, Corollary 1a implies that increasing the number of source examples is always beneficial for transfer learning when the source and target tasks are related ($\varepsilon \approx 0$), which matches intuition. Next, Corollary 1b implies that increasing the number of source classes while leaving the number of source examples fixed can decrease performance (i.e. if $2S - 1 - ST > 0$), even for similar source and target tasks satisfying $\varepsilon \approx 0$. This matches the experiments in Fig. 2c, where we observed that increasing the number of source classes when keeping the number of source examples fixed can be detrimental to the performance. This is intuitive for transferring from ImageNet32 to CIFAR10, since we would be adding classes that are not as useful for predicting objects in CIFAR10. However, note that such behavior is a priori unexpected given generalization bounds for multi-task learning problems[31–33], which show that increasing the number of tasks decreases the overall risk. Our non-asymptotic analysis demonstrates that such decrease in risk only holds as the number of classes *and* the number of examples per class increase. Corollary 1c implies that when the source and target task are similar and the number of source classes is less than the data dimension, transfer learning with the projected predictor is always better than training only on the target task. Moreover, if the number of source classes is finite ($C = 0$), Corollary 1c implies that the risk of the projected predictor decreases an order of magnitude faster than the baseline predictor. In particular, the risk of the baseline predictor is given by $(1 - T)\|\omega_t\|^2$, while that of the projected predictor is given by $(1 - T)^2 \|\omega_t\|^2$. Note also that when the number of target samples is small relative to the dimension, Corollary 1c implies that decreasing the number of source classes has minimal effect on the risk. Lastly, Corollary 1d implies that when $T$ and $C$ are small, the risk of the projected predictor is roughly that of a baseline predictor trained on twice the number of samples.

We derive scaling laws for translated predictors in the linear setting. Analogously to the case for projection, we analyze the risk of the translated predictor when $\hat{\omega}_s$ is the minimum norm solution to $\|y_s - \omega X_s\|_F^2$ and $\hat{\omega}_c$ is the minimum norm solution to $\|y_t - \hat{\omega}_s X_t - \omega X_t\|_F^2$.

**Theorem 2.** Let $\mathcal{X} = \mathbb{R}^d$, $\mathcal{Y}_s = \mathbb{R}^{c_s}$, $\mathcal{Y}_t = \mathbb{R}^{c_t}$, and let $\hat{\omega}_t = \hat{\omega}_s + \hat{\omega}_c$ where $\hat{\omega}_s = y_s X_s^\dagger$ and $\hat{\omega}_c = (y_t - \hat{\omega}_s X_t) X_t^\dagger$. Assuming that $\mathbb{P}_s$ and $\mathbb{P}_t$ are independent, isotropic distributions on $\mathbb{R}^d$, then the risk $\mathcal{R}(\hat{\omega}_t)$ is given by

$$\mathcal{R}(\hat{\omega}_t) = \left[ \frac{\|\omega_s - \omega_t\|_F^2}{\|\omega_t\|_F^2} + \left(1 - \frac{n_s}{d}\right)\left(1 - \frac{\|\omega_s - \omega_t\|_F^2}{\|\omega_t\|_F^2}\right) \right] \mathcal{R}(\hat{\omega}_b), \tag{8}$$

where $\hat{\omega}_b = y_t X_t^\dagger$ is the baseline predictor.

The proof is given in SI Note 5. Theorem 2 formalizes several intuitions regarding when translation is beneficial. In particular, we first observe that if the source model $\omega_s$ is recovered exactly (i.e. $n_s = d$), then the risk of the translated predictor is governed by the distance between the oracle source model and target model, i.e., $\|\omega_s - \omega_t\|$. Hence, the translated predictor generalizes better than the baseline predictor if the source and target models are similar. In particular, by flattening the matrices $\omega_s$ and $\omega_t$ into vectors and assuming $\|\omega_s\| = \|\omega_t\|$, the translated predictor outperforms the baseline predictor if the angle between the flattened $\omega_s$ and $\omega_t$ is less than $\frac{\pi}{4}$. On the other hand, when there are no source samples, the translated predictor is exactly the baseline predictor and the corresponding risks are equivalent. In general, we observe that the risk of the translated predictor is simply a weighted average between the baseline risk and the risk in which the source model is recovered exactly.

Comparing Theorem 2 to Theorem 1, we note that the projected predictor and the translated predictor generalize based on different

quantities. In particular, in the case when $n_s = d$, the risk of the translated predictor is a constant multiple of the baseline risk while the risk of the projected predictor is a multiple of the baseline risk that decreases with $n_t$. Hence, depending on the distance between $\omega_s$ and $\omega_t$, the translated predictor can outperform the projected predictor or vice-versa. As a simple example, consider the setting where $\omega_s = \omega_t$, $n_s = d$, and $n_t, c_s < d$; then the translated predictor achieves 0 risk while the projected predictor achieves non-zero risk. When $\mathcal{Y}_s = \mathcal{X}_t$, we suggest combining the projected and translated predictors, as we did in the case of virtual drug screening. Otherwise, our results suggest using the translated predictor for transfer learning problems involving distribution shift in the features but no difference in the label sets, and the projected predictor otherwise.

## Discussion

In this work, we developed a framework that enables transfer learning with kernel methods. In particular, we introduced the projection and translation operations to adjust the predictions of a source model to a specific target task: While projection involves applying a map directly to the predictions given by the source model, translation involves adding a map to the predictions of a source model. We demonstrated the effectiveness of the transfer learned kernels on image classification and virtual drug screening tasks. Namely, we showed that transfer learning increased the performance of kernel-based image classifiers by up to 10% over training such models directly on the target task. Interestingly, we found that transfer-learned convolutional kernels performed comparably to transfer learning using the corresponding finite-width convolutional networks. In virtual drug screening, we demonstrated that the transferred kernel methods provided an improvement over prior work[12], even in settings where none of the target drug and cell lines were present in the source task. For both applications, we analyzed the performance of the transferred kernel model as a function of the number of target examples and observed empiricallly that the transferred kernel followed a simple logarithmic trend, thereby enabling predicting the benefit of collecting more target examples on model performance. Lastly, we mathematically derived the scaling laws in the linear setting, thereby providing a theoretical foundation for the empirical observations. We end by discussing various consequences as well as future research directions motivated by our work.

### Benefit of pretraining kernel methods on large datasets

A key contribution of our work is enabling kernels trained on large datasets to be transferred to a variety of downstream tasks. As is the case for neural networks, this allows pre-trained kernel models to be saved and shared with downstream users to improve their applications of interest. A key next step to making these models easier to save and share is to reduce their reliance on storing the entire training set, such as by using coresets[60]. We envision that by using such techniques in conjunction with modern advances in kernel methods, the memory and runtime costs could be drastically reduced.

### Reducing kernel evaluation time for state-of-the-art convolutional kernels

In this work, we demonstrated that it is possible to train convolutional kernel methods on datasets with over 1 million images. In order to train such models, we resorted to using the CNTK of convolutional networks with a fully connected last layer. While other architectures, such as the CNTK of convolutional networks with a global average pooling last layer, have been shown to achieve superior performance on CIFAR10[14], training such kernels on 50k images from CIFAR10 is estimated to take 1200 GPU hours[61], which is more than three orders of magnitude slower than the kernels used in this work. The main computational bottleneck for using such improved convolution kernels is evaluating the kernel function itself. Thus an important problem is to improve the computation time for such kernels, which would allow training better convolutional kernels on large-scale image datasets, which could then be transferred using our framework to improve the performance on a variety of downstream tasks.

### Using kernel methods to adapt to distribution shifts

Our work demonstrates that kernels pre-trained on a source task can adapt to a target task with distribution shift when given even just a few target training samples. This opens novel avenues for applying kernel methods to tackle distribution shift in a variety of domains, including healthcare or genomics in which models need to be adapted to handle shifts in cell lines, populations, batches, etc. In the context of virtual drug screening, we showed that our transfer learning approach could be used to generalize to new cell lines. The scaling laws described in this work may provide an interesting avenue to understand how many samples are required in the target domain for more complex domain shifts, such as from a model organism like mouse to humans, a problem of great interest in the pharmacological industry.

## Methods

### Overview of image classification datasets

For projection, we used ImageNet32 as the source dataset and CIFAR10, Oxford 102 Flowers, DTD, and a subset of SVHN as the target datasets. For all target datasets, we used the training and test splits given by the PyTorch library[62]. For ImageNet32, we used the training and test splits provided by the authors[24]. An overview of the number of training and test samples used from each of these datasets is outlined below.

1. **ImageNet32** contains $1,281,167$ training images across 1000 classes and 50k images for validation. All images are of size $32 \times 32 \times 3$.
2. **CIFAR10** contains 50k training images across 10 classes and 10k images for validation. All images are of size $32 \times 32 \times 3$.
3. **Oxford 102 Flowers** contains 1020 training images across 102 classes and 6149 images for validation. Images were resized to $32 \times 32 \times 3$ for the experiments.
4. **DTD** contains 1880 training images across 47 classes and 1880 images for validation. Images were resized to size $32 \times 32 \times 3$ for experiments.
5. **SVHN** contains 73257 training images across 10 classes and 26302 images for validation. All images are of size $32 \times 32 \times 3$. In Fig. 2, we used the same 500 training image subset for all experiments.

### Training and architecture details

**Model descriptions.**

1. **Laplace Kernel:** For samples $x, \tilde{x}$, and bandwidth parameter $L$, the kernel is of the form:

$$\exp\left(-\frac{\|x - \tilde{x}\|_2}{L}\right).$$

For our experiments, we used a bandwidth of $L = 10$ as in[63], selected through cross-validation.

2. **NTK:** We used the NTK corresponding to an infinite width ReLU fully connected network with 5 hidden layers. We chose this depth as it gave superior performance on image classification task considered in[64].

3. **CNTK:** We used the CNTK corresponding to an infinite width ReLU convolutional network with 6 convolutional layers followed by a fully connected layer. All convolutional layers used filters of size $3 \times 3$. The first 5 convolutional layers used a stride size of 2 to downsample the image representations. All convolutional layers used zero padding. The CNTK was computed using the Neural Tangents library[61].

4. **CNN:** We compare the CNTK to a finite-width CNN of the same architecture that has 16 filters in the first layer, 32 filters in the second layer, 64 filters in the third layer, 128 filters in the fourth layer, and 256 filters in the fifth and sixth layers. In all experiments, the CNN was trained using Adam with a learning rate of $10^{-4}$. Our choice of learning rate is based on its effectiveness in prior works[65,66].

**Details for projection experiments.** For all kernels trained on ImageNet32, we used EigenPro[29]. For all models, we trained until the training accuracy was greater than 99%, which was at most 6 epochs of EigenPro. For transfer learning to CIFAR10, Oxford 102 Flowers, DTD, and SVHN, we applied a Laplace kernel to the outputs of the trained source model. For CIFAR10 and DTD, we solved the kernel regression exactly using NumPy[67]. For DTD and SVHN, we used ridge regularization with a coefficient of $10^{-4}$ to avoid numerical issues with solving exactly. The CNN was trained for at most 500 epochs on ImageNet32, and the transferred model corresponded to the one with highest validation accuracy during this time. When transfer learning, we fine-tuned all layers of the CNN for up to 200 epochs (again selecting the model with the highest validation accuracy on the target task).

**Details for translation experiments.** For transferring kernels from CIFAR10 to CIFAR-C, we simply solved kernel regression exactly (no ridge regularization term). For the corresponding CNNs, we trained the source models on CIFAR10 for 100 epochs and selected the model with the best validation performance. When transferring CNNs to CIFAR-C, we fine-tuned all layers of the CNN for 200 epochs and selected the model with the best validation accuracy. When translating kernels from ImageNet32 to CIFAR10 in SI Fig. S5, we used the following aggregated class indices in ImageNet32 to match the classes in CIFAR10:

1. plane = {372, 230, 231, 232}
2. car = {265, 266, 267, 268 }
3. bird = {383, 384, 385, 386}
4. cat = {8, 10, 11, 55}
5. deer = {12, 9, 57}
6. dog = {131, 132, 133, 134}
7. frog = {499, 500, 501, 494}
8. horse = {80, 39}
9. ship = {243, 246, 247, 235}
10. truck = {279, 280, 281, 282}.

**Details for virtual drug screening.** We used the NTK corresponding to a 1 hidden layer ReLU fully connected network with an offset term. The same model was used in[12]. We solved kernel ridge regression when training the source models, baseline models, and transferred models. For the source model, we used ridge regularization with a coefficient of 1000. To select this ridge term, we used a grid search over {1, 10, 100, 1000, 10000} on a random subset of 10k samples from the source data. We used a ridge term of 1000 when transferring the source model to the target data and a term of 100 when training the baseline model. We again tuned the ridge parameter for these models over the same set of values but on a random subset of 1000 examples for one cell line (A549) from the target data. We used 5-fold cross validation for the target task and reported the metrics computed across all folds.

**Comparison with multiple kernel learning approaches.** In SI Fig. S7, we compare our translation approach to the approach of directly using multiple kernel learning algorithms such as Centered Kernel Alignment (CKA)[68], EasyMKL[69], FHeuristic[70], and Proportionally Weighted Multiple Kernels (PWMK)[71] to learn a kernel on the source task and train the target model using the learned kernel. Due to the computational limitations of these prior methods, we only consider the subtask of classifying cars and deer in CIFAR10 (source) and transferring to the 19 corruptions in CIFAR10-C (target). The source task contains $10, 000$ training samples and 2000 test samples, while the 19 target tasks each contain 1000 training samples and 1000 test samples. The multiple kernel learning algorithms learn combinations of a Laplace kernel with bandwidth 10, a Gaussian kernel of the form $K_G(x, z) = \exp(-\gamma \| x - z \|^2)$ with $\gamma = 0.001$, and the linear kernel $K_L(x, z) = x^T z$. We choose the bandwidth for the Laplace kernel from[29] and the value of $\gamma$ for the Gaussian kernel so that the entries of the Gaussian kernel are on the same order of magnitude as the Laplace kernel for this task. For each kernel learning algorithm, we train a source model and then compare the following three models on the target task: (1) the baseline model in which we use the kernel learning algorithm directly on the target task; (2) the transfer learned kernel model in which we use the weights from the source task to combine the kernels on the target task; and (3) translating the learned kernel on the source task using our translation methodology. As shown in SI Fig. S7, the transfer learned kernel outperforms the baseline kernel for almost all multiple kernel learning algorithms (except FHeuristic), and it is outperformed by our translation methodology in all cases.

**Remark.** We presented a comparison on this simple binary classification task for the following computational reasons. First, we considered binary classification, since prior multiple kernel learning approaches implemented in MKLPy scale poorly to multi-class problems. While there is no computational price to be paid for our method for multi-class classification, prior methods build one kernel per class and thus require 10 times more compute and memory when using all 10 classes in CIFAR10. Secondly, we compared with only translation and not projection since prior multiple kernel learning methods scale poorly to the ImageNet32 dataset used in our projection experiments. Namely, multiple kernel learning methods require materializing the kernel matrix, which for ImageNet32 would take up more than 3.5 terabytes of memory as compared to 128 gigabytes for our method.

## Projection scaling laws

For the curves showing the performance of the projected predictor as a function of the number of target examples in Fig. 2b and SI Fig. S2a, b, we performed a scaling law analysis. In particular, we used linear regression to fit the coefficients $a, b$ of the function $y = a \log_2 x + b$ to the points from each of the curves presented in the figures. Each curve in these figures has 50 evenly spaced points and all accuracies are averaged over 3 seeds at each point. The $R^2$ values for each of the fits is presented in SI Fig. S4. Overall, we observe that all values are above 0.944 and are higher than 0.99 for CIFAR10 and SVHN, which have more than 2000 target training samples. Moreover, by fitting the same function on the first 5 points from these curves for CIFAR10, we are able to predict the accuracy on the last point of the curve within 2% of the reported accuracy.

## Preprocessing for CMAP data

While CMAP contains 978 landmark genes, we removed all genes that were 1 upon $\log_2(x + 1)$ scaling the data. This eliminates 135 genes and removes batch effects identified in[50] for each cell line. Following the methodology of[50], we also removed all perturbations with dose less than 0 and used only the perturbations that had an associated simplified molecular-input line-entry system (SMILES) string, which resulted in a total of 20, 336 perturbations. Following[50], for each of the 116, 228 observed drug and cell type combinations we then averaged the gene expression over all the replicates.

## Metrics for evaluating virtual drug screening

Let $\hat{y} \in \mathbb{R}^{n \times d}$ denote the predicted gene expression vectors and let $y^* \in \mathbb{R}^{n \times d}$ denote the ground truth. Let $\bar{y}^{(i)} = \frac{1}{d} \sum_{j=1}^{d} y_j^{(i)}$. Let $\hat{y}_v, y_v^* \in \mathbb{R}^{dn}$ denote vectorized versions of $\hat{y}$ and $y^*$. We use the same three

metrics as those considered in[12,51]. All evaluation metrics have a maximum value of 1 and are defined below.

1. *Pearson r value*:

$$r = \frac{\langle \hat{y}_v, y_v^* \rangle}{\|\hat{y}_v\|_2 \|y_v^*\|_2}.$$

2. *Mean $R^2$*:

$$R^2 = \frac{1}{n} \sum_{i=1}^{n} \left( 1 - \frac{\sum_{j=1}^{d} (\hat{y}_j^{(i)} - y_j^{*(i)})^2}{\sum_{j=1}^{d} (y_j^{*(i)} - \bar{y}^{(i)})^2} \right).$$

3. *Mean Cosine Similarity*:

$$c = \frac{1}{n} \sum_{i=1}^{n} \frac{\langle \hat{y}^{(i)}, y^{*(i)} \rangle}{\|\hat{y}^{(i)}\|_2 \|y^{*(i)}\|_2}.$$

We additionally subtract out the mean over cell type before computing cosine similarity to avoid inflated cosine similarity arising from points far from the origin.

### DepMap analysis

To provide another application of our framework in the context of virtual drug screening, we used projection to transfer the kernel methods trained on imputing gene expression vectors in CMAP to predicting the viability of a drug and cell line combination in DepMap[53]. Viability scores in DepMap are real values indicating how lethal a drug is for a given cancer cell line (negative viability indicates cell death). To transfer from CMAP to DepMap, we trained a kernel method to predict the gene expression vectors for 55,462 cell line and drug combinations for the 64 cell lines from CMAP that do not overlap with DepMap. We then used projection to transfer the model to the 6 held-out cell lines present in both CMAP and DepMap, which are PC3, MCF7, A375, A549, HT29, and HEPG2. Analogously to our analysis of CMAP, we stratified the target dataset by drugs that appear in both the source and target tasks (9726 target samples) and drugs that are only found in the target task but not in the source task (2685 target samples). For this application, we found that Mol2Vec[72] embeddings of drugs outperformed 1024-bit circular fingerprints. We again used a 1-hidden layer ReLU NTK with an offset term for this analysis and solved kernel ridge regression with a ridge coefficient of 100.

SI Fig. S9a shows the performance of the projected predictor as a function of the number of target samples when transferring to a target task with drugs that appear in the source task. All results are averaged over 5 folds of cross-validation and across 5 random seeds for the subset of target samples considered in each fold. It is apparent that performance is greatly improved when there are fewer than 2000 samples, thereby highlighting the benefit of the imputed gene expression vectors in this setting. Interestingly, as in all the previous experiments, we find a clear logarithmic scaling law: fitting the coefficients of the curve $y = a \log_2 x + b$ to the 76 points on the graph yields an $R^2$ of 0.994, and fitting the curve to the first 10 points lets us predict the $R^2$ for the last point on the curve within 0.03. SI Fig. S9b shows how the performance on the target task is affected by the number of genes predicted in the source task. Again performance is averaged over 5 fold cross-validation and across 5 seeds per fold. When transferring to drugs that were available in the source task, performance monotonically increases when predicting more genes. On the other hand, when transferring to drugs that were not available in the target task, performance begins to degrade when increasing the number of predicted genes. This is intuitive, since not all genes would be useful for predicting the effect of an unseen drug and could add noise to the prediction problem upon transfer learning.

### Hardware details

All experiments were run using two servers. One server had 128GB of CPU random access memory (RAM) and 2 NVIDIA Titan XP GPUs each with 12GB of memory. This server was used for the virtual drug screening experiments and for training the CNTK on ImageNet32. The second server had 128GB of CPU RAM and 4 NVIDIA Titan RTX GPUs each with 24GB of memory. This server was used for all the remaining experiments.

### Data availability

All datasets considered in this work are publicly available. The standard image classification datasets considered in this work are available directly through the PyTorch library[62]. CMap data is available through the following website https://www.ncbi.nlm.nih.gov/geo/query/acc.cgi?acc=GSE92742, and we used the level 2 data given in the file GSE92742_Broad_LINCS_Level2_GEX_epsilon_n1269922x978.gctx. DepMap data is available through the following website https://depmap.org/repurposing/, and we used the primary screen data.

### Code availability

All code is available at https://github.com/uhlerlab/kernel_tf[73].

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

## Acknowledgements

The authors were partially supported by NCCIH/NIH (1DP2AT012345), NSF (DMS-1651995), ONR (N00014-22-1-2116), the MIT-IBM Watson AI Lab, AstraZeneca, the Eric and Wendy Schmidt Center at the Broad Institute, and a Simons Investigator Award (to C.U.).

## Author contributions

A.R., M.R.L, C.U. designed the research, A.R., M.R.L., N.P. developed and implemented the algorithms, all authors performed model and data analysis. A.R., M.R.L., C.U. wrote the paper.

## Competing interests

The authors declare no competing interests.
