## [Peer Review File · Nature Communications]

REVIEWER COMMENTS

Reviewer #2 (Remarks to the Author):

The paper proposes a general framework for transfer learning with kernel methods. The major claim is that, unlike transfer learning with deep neural networks, it has been unclear how to perform transfer learning for kernel methods. Given a kernel predictor trained on the source data set, the paper proposes two simple operations, namely projection, and translation, to "transfer" the learned predictor to the target data set. The evaluations are conducted on standard benchmark datasets including ImageNet32, CIFAR10, Oxford 102 Flower, Describable Textures, and SVHN datasets. On these datasets, the paper reports that the proposed methods can achieve up to 10% improvement over the baseline method (i.e., applying the predictor previously trained on the source data set directly on the target data set). The proposed method also shows superior results on virtual drug screening tasks compared to the state-of-the-art kernel method based on the neural tangent kernel (NTK). Lastly, the paper supplements these empirical finds with insightful theoretical results under the linearity assumption.

The noteworthy results of the work are twofold. First, the proposed methods (projection and translation) are simple and can be applied to other learning algorithms in addition to kernel methods. Second, the empirical results are promising which demonstrates the effectiveness of the proposed methods on the standard benchmark data sets and at least one real-world application in virtual drug screening.

On the other hand, the weaknesses of this work are threefold. First, it lacks a careful literature review in the area of transfer learning with kernel methods. I disagree with the claim that "it has been unclear how to perform transfer learning for kernel methods" (abstract) and that "developing an algorithm for transfer learning with kernel methods for general source and target tasks has been an open problem" (Introduction). On the contrary, transfer learning has been studied extensively in the kernel methods community. This comes under different names including multi-task learning [1,2], domain adaptation (DA) [3,4], and domain generalization (DG) [5,6], among others. Kernel methods have been applied in all of these areas. Hence, I believe the paper mischaracterizes existing literature in transfer learning with kernel methods. Compared to deep learning, it is true that there is no such thing as a "pre-trained" kernel in the literature, but this can already be achieved by existing kernel learning methods, e.g., [7]; see also my comments/questions below.

Second, there is no clear specific motivation for the proposed "projection" and "translation" operations. What motivated the authors to consider these two operations specifically for transfer learning with kernel methods? They are natural and somewhat straightforward operations that anyone could come up with in the first place. Without a clear motivation, the proposed methods appear quite ad-hoc. Furthermore, as stated on page 4 that "Definition 1 is applicable to any machine learning method, [...]",

it looks like the proposed "projection" and "translation" operations may not have anything specific to do with kernel methods. What I expect to see in these two operations is some sort of technical innovation specific to kernel methods. Otherwise, what kind of gains or benefits would the kernel methods bring in this context?

Lastly, while the theoretical results are insightful, they rely on the "linearity" assumption which I believe is too simplistic in this setting. The paper claims that these theoretical results explain the empirical findings well, but I have some doubts about this; see my comments/questions below.

To sum up, the paper proposes two simple operations for transfer learning with kernel methods that show promising empirical results. However, the paper lacks well-articulated motivation and principles behind the proposed operations. It also remains unclear what we gain by focusing on kernel methods. The theoretical results are insightful but rely on an unrealistic linearity assumption.

[1] Kernels for Multi-task Learning, NeurIPS 2004

[2] Learning Multiple Tasks with Kernel Methods, JMLR 2005

[3] Covariate Shift by Kernel Mean Matching, Book Chapter

[4] Domain Adaptation via Transfer Component Analysis, IJCAI 2009

[5] Generalizing from Several Related Classification Tasks to a New Unlabeled Sample, NeurIPS 2011

[6] Domain Generalization via Invariant Feature Representation, ICML 2013

[7] Multiple Kernel Learning Algorithms, JMLR 2011

===

Some questions and suggestions:

- First paragraph of Introduction: "However, developing an algorithm for transfer learning with kernel methods for general source and target tasks has been an open problem." In what sense is this an open problem? For example, transfer learning with kernel methods has been studied extensively in the domain adaptation (DA) literature. Also, since for kernel methods, we can write $f(x)$ as $f(x) = w.T\phi(x) = w.T k(x,.)$, one can consider the following baseline approach: First, we train both w and kernel $k(.,.)$ on the source data set using "kernel learning" procedure. Then, we re-learn only the predictor w on the target data set while keeping the learned kernel fixed. This simple baseline can indeed be viewed as a transfer learning with kernels and has been adopted in the past. How does it compare with the proposed method?

- Caption of Figure 1: Typo, "cell linees"  "cell lines"

- Page 2: "[...] translate to adding a new layer to the end of a neural network." This is unclear. Can the authors elaborate?

- In the introduction, it is unclear what motivated the authors to consider specifically projection and translation for transfer learning. Can the authors elaborate at least at an intuition level?

- Definition 1: What is the motivation and/or principle behind this? What makes this operation specific to kernel methods? What kind of benefits does it bring by focusing on kernel methods?

- In the first set of experiments on ImageNet-32, did the authors already mention somewhere that the training/validation data in ImageNet-32 do not leak into the target datasets, e.g., CIFAR10, Oxford 102 Flower, etc. Otherwise, the reported results would be problematic.

- Figure 2b (ImageNet-32 to Flower-102): Why does the gap in accuracy between transferred kernels baselines appear to widen as the number of target examples increases? This is counterintuitive as this gap should instead reduce as we observe more target examples.

- Definition 2: Again, what makes this operation special for kernel methods? What kind of benefits does it bring by focusing on kernel methods?

- Page 6: "We note that training translated predictors is also a new form of boosting." Can the authors elaborate?

- Page 7: "Overall, our results showcase that translation is effective at adapting kernel methods to distribution shift in image classification." The conclusion is specific to kernel methods, while it remains unclear what makes Definition 2 special for kernel methods.

- Page 9: "[...] the concatenation above, we get the translated predictor." How? I tried, but I cannot see how we can get the translated predictor because \hat{f}_s disappears from the equation.

- Page 9: "[...] the performance gap between the transferred NTK and the baseline NTK grows as more target examples are collected." Can the authors elaborate more on this behavior? This is related to one of my previous questions.

- Page 9: Typo, "[...] for other transfer learning tasts [...]"  "[...] for other transfer learning tasks [...]"

- Theoretical analysis: Why did the authors make the linearity assumption? It is highly unrealistic in the context of this work. Do the analyses make sense for classification problems or the focus is on regression problems?

- Theorem 1: How does this result explain the "logarithmic" scaling law observed in the empirical results? Where does the logarithmic dependency appear in the theoretical result?

- Page 10, Remark: "[...] corroborate surprising dependencies [...]" Why is this surprising?

- Theorem 2: Typo, "the the"  "the"

- Page 11: "The proof is given in SI Section E" The period is missing at the end of the sentence.

- Page 11, Discussion: Typo, "empirtically"  "empirically", "drastially"  "drastically"

- Page 12: "[...] we used a bandwidth of $L=10$." How did the authors choose this hyperparameter? Did the authors perform model selection?

- Page 13: "[...] the CNN was trained using Adam with a learning rate of 10^{-4} ." How did the authors choose this hyperparameter?

- Page 13: Would the proposed methods still applicable if we do not have access to these aggregated class indices in practice?

- Caption of Figure 9: Typo, "[...] examples per class increases.."  "[...] examples per class increases."

Reviewer #3 (Remarks to the Author):

The paper is on an important topic - transfer learning - and the authors have done very good work both theoretically and experimentally. The experimental results shown are also strong. They consider two types of transfer learning - with the source and target tasks having the same or different label sets.

I have two major concerns, hence recommending a major (risky) revision.

First, the authors unfortunately partly “reinvent the wheel”. While they do not refer to earlier work, some of the ideas they propose (e.g. the additive approach for the case of source and target tasks having the same labels - definition 2, page 6) is very similar to work proposed since the early 2000s - see these articles and references therein (also to earlier work on multi-task and transfer learning):

T. Evgeniou, M. Pontil. Regularized multi--task learning. Proceedings of the tenth ACM SIGKDD international conference on Knowledge, 2004.

T. Evgeniou, C.A. Micchelli, M. Pontil. Learning multiple tasks with kernel methods. Journal of Machine Learning Research, 2005

A. Argyriou, T. Evgeniou, M. Pontil. Convex multi-task feature learning. Machine Learning 73 (3), 243-272, 2008.

Tommasi, T., Orabona, F., and Caputo, B. Safety in Numbers: Learning Categories from Few Examples with Multi Model Knowledge Transfer. CVPR 2010.

The authors should compare with - and build upon - these series of works. It is not clear how major are the proposed innovations relative to that - and other - work, hence the recommendation is for a major revision (not a rejection - hoping the authors will provide significant advances in this important topic).

Second, note that there are also theoretical results that the authors can compare with/build upon, such as:

J Baxter

A model of inductive bias learning

J. Artif. Intell. Res. (JAIR) 12, 149-198, 2000.

Ando, Rie Kubota, Tong Zhang, and Peter Bartlett.

A framework for learning predictive structures from multiple tasks and unlabeled data." Journal of Machine Learning Research 6, no. 11 (2005).

Maurer A, Pontil M, Romera-Paredes B. The benefit of multitask representation learning. Journal of Machine Learning Research. 2016;17(81):1-32.

I Kuzborskij, F Orabona

Fast Rates by Transferring from Auxiliary Hypotheses

Machine Learning, 2017

Denevi, C Ciliberto, D Stamos, M Pontil

Learning to learn around a common mean

Advances in Neural Information Processing Systems 31, 2018

M Khodak, MFF Balcan, AS Talwalkar

Adaptive gradient-based meta-learning methods

Advances in Neural Information Processing Systems 32, 2019

Minor: Theorem 2 in p. 10 has some typos ("the" is "then", and I assume w_b is w_s ? But then can we also say something about w_s ?)

Reviewer # 2 response:

The paper proposes a general framework for transfer learning with kernel methods. The major claim is that, unlike transfer learning with deep neural networks, it has been unclear how to perform transfer learning for kernel methods. Given a kernel predictor trained on the source data set, the paper proposes two simple operations, namely projection, and translation, to "transfer" the learned predictor to the target data set. The evaluations are conducted on standard benchmark datasets including ImageNet32, CIFAR10, Oxford 102 Flower, Describable Textures, and SVHN datasets. On these datasets, the paper reports that the proposed methods can achieve up to 10% improvement over the baseline method (i.e., applying the predictor previously trained on the source data set directly on the target data set). The proposed method also shows superior results on virtual drug screening tasks compared to the state-of-the-art kernel method based on the neural tangent kernel (NTK). Lastly, the paper supplements these empirical finds with insightful theoretical results under the linearity assumption.

The noteworthy results of the work are twofold. First, the proposed methods (projection and translation) are simple and can be applied to other learning algorithms in addition to kernel methods. Second, the empirical results are promising which demonstrates the effectiveness of the proposed methods on the standard benchmark data sets and at least one real-world application in virtual drug screening.

We thank the reviewer for their thoughtful and detailed feedback on our paper. We are glad that the reviewer found our proposed methodology widely applicable and our empirical results to demonstrate the effectiveness of our method. We hope our point-by-point response together with a careful literature review detailing the conceptual differences between prior works and our work as well as additional experimental comparisons (see Fig. 1) address the issues raised by the reviewer below.

On the other hand, the weaknesses of this work are threefold.

1. First, it lacks a careful literature review in the area of transfer learning with kernel methods. I disagree with the claim that "it has been unclear how to perform transfer learning for kernel methods" (abstract) and that "developing an algorithm for transfer learning with kernel methods for general source and target tasks has been an open problem" (Introduction). On the contrary, transfer learning has been studied extensively in the kernel methods community. This comes under different names including multi-task learning [1,2], domain adaptation (DA) [3,4], and domain generalization (DG) [5,6], among others. Kernel methods have been applied in all of these areas. Hence, I believe the paper mischaracterizes existing literature in transfer learning with kernel methods. Compared to deep learning, it is true that there is no such thing as a "pre-trained" kernel in the literature, but this can already be achieved by existing kernel learning methods, e.g., [7]; see also my comments/questions below.

We thank the reviewer for pointing out these related works. We would like to clarify that while these works provide methods for improving kernels for multi-class problems and for domain adaptation tasks, they do not provide methods for transfer learning between general source and target tasks with possibly differing label dimensions, the problem considered in our work. Moreover, these prior methods are unable to scale computationally to several of the tasks considered in our work. In particular, the works [3, 4, 6, 7] mentioned by the reviewer require materializing the full kernel matrix, which for kernels trained on ImageNet32 would take at least 3.5 terabytes of memory (as compared to 128 gigabytes for our method). We briefly summarize each of the suggested related works below and describe the conceptual differences to our work. This lack of scalable methods to transfer between general source and target tasks with possibly differing label dimension is what we meant in the abstract and introduction. To clarify this, we edited the sentences referenced by the reviewer in the abstract and introduction as follows:

Abstract: "While kernel methods are conceptually and computationally simple machine learning models that are competitive on a variety of tasks, it has been unclear how to develop scalable kernel-based transfer learning methods across general source and target tasks with possibly differing label dimensions."

Introduction: "Yet, developing scalable algorithms for transfer learning with kernel methods for general source and target tasks with possibly differing label dimensions has been an open problem. [...] In this work, we present a general, scalable framework for performing transfer learning with kernel methods. [...] Our formulation of these operations makes them compatible with recent pre-conditioned gradient descent kernel regression solvers such as EigenPro [1], thereby allowing our framework to easily scale to datasets such as ImageNet32 with over one million samples."

In the following, we describe in more detail the contributions of our work in the context of the prior

works mentioned by the reviewer. In particular, we note that:

- (1) the works [3, 4, 6, 7] require materializing a full kernel matrix, which is computationally infeasible on our hardware for the tasks involving ImageNet32 considered in our work;
- (2) the works [1, 2] involve computing matrix-valued kernels, which for ImageNet32 or virtual drug screening data from CMap would require over 10^6 times more compute than our current methodology (e.g., our ImageNet32 experiment with the CNTK would take well over a year on a cloud instead of days on a single server);
- (3) the works [5, 6] provide methodology for settings in which the source and target datasets have the same label dimension, whereas our projection method can handle tasks with differing label dimension.

In the following, we provide a more detailed description of each of the works and highlight the conceptual differences to our proposed method.

- [1] *Kernels for Multi-task Learning*, NeurIPS 2004 (reference [2] in the list below)

This theoretical work establishes a foundation for multi-task learning using matrix-valued kernels. The authors characterize linear matrix-valued kernels and present an extension to nonlinear matrix-valued kernels by using products of kernel functions. We note that matrix-valued kernels are computationally expensive, requiring $c^2 \times n^2$ computations for c labels and n data points. For the case of ImageNet32 where $c = 10^3$ and $n \approx 10^6$, this method would require computing 10^6 more values than our current method. Even when computing the kernel in batches using preconditioned mini-batch gradient descent solvers such as EigenPro, this would increase runtime for the CNTK on ImageNet32 from a few days to well over a year. This prior work also requires choosing various hyperparameters (matrices denoted B_ℓ in the paper in addition to the bandwidth of the kernel). Furthermore, this approach requires knowledge of all target tasks at the time of training the source model, which is not required in our work.

- [2] *Learning Multiple Tasks with Kernel Methods*, JMLR 2005 (reference [3] in the list below)

This work investigates learning multiple related tasks simultaneously using kernel methods and regularization by extending single-task kernel methods to multi-task learning. The authors develop a family of multi-task kernel functions to model task relationships and demonstrate improved performance over standard single-task learning in cases with numerous related tasks and limited data per task. Since this paper employs the same method as in [1], as seen by comparing equation 11 to equation 4.14 from [1], the computational limitations of [1] apply also to this work. While this work addresses the need for expert choice of the matrices B_ℓ in [1], this comes at the cost of iterating the computationally expensive training process multiple times.

- [3] *Covariate Shift by Kernel Mean Matching*, Book Chapter (reference [4] in the list below)

This work presents a method for re-weighting training data to match the distribution of test data using kernel mean matching (KMM). Unlike the general transfer learning tasks in our work where source and target labels may be different, this work focuses on the specific problem of distribution shift when the labels between the source and target task are given by the same set. This work makes the following assumptions, which need not hold in our work: (1) $\mathbb{P}(y|x)$ is unchanged across source and target task (see Assumption 1.1 of [3]); and (2) the support of the test distribution lies in the support of the training distribution. In addition, the method described in this work requires solving a quadratic program, which involves materializing the full kernel matrix. For large datasets such as ImageNet32, materializing the full matrix would require over 3.5 terabytes of memory, which is more than available on our hardware or required for our method (128 gigabytes).

- [4] *Domain Adaptation via Transfer Component Analysis*, IJCAI 2009 (reference [5] in the list below)

This work presents transfer component analysis (TCA), a method for domain adaptation that finds a feature representation minimizing the Maximum Mean Discrepancy (MMD) between the empirical source and target datasets in a Reproducing Kernel Hilbert Space (RKHS). This work is focused on the same setting as in [3] and contains assumptions not required in our work such as the assumption that $\mathbb{P}(y|x)$ is unchanged across source and target task. Other limitations of this work are its memory requirements (already discussed in [3] above, since the method requires materializing the full kernel matrix) and computational requirements. As stated by the authors, solving the proposed optimization

problems in Eqs. (9) and (10) takes $O(m(n_1 + n_2)^2)$ where m is a hyper-parameter denoting the rank of the projection matrix W used in their work and n_1, n_2 denote the number of source and target examples respectively. The quadratic dependency on $n_1 + n_2$ makes the procedure an order of magnitude slower than our methodology using EigenPro, which is problematic for datasets such as ImageNet32 and CIFAR10 considered in our work.

[5] Generalizing from Several Related Classification Tasks to a New Unlabeled Sample, NeurIPS 2011 (reference 6 in the list below)

This work proposes a kernel-based approach for predicting class labels on an unlabeled test dataset by leveraging several related training datasets. The authors introduce kernel regression on the input space (\hat{P}_X, x) and propose kernels of the form $\bar{k}((P_1, x_1), (P_2, x_2)) = k_P(P_1, P_2)k_X(x_1, x_2)$. In contrast to the multi-class problems considered in our work, this work focuses on the binary classification setting and constructs a kernel with $(\sum_{i=1}^N n_i)^2$ entries, where N denotes the number of source datasets, with n_i examples in each dataset. This work does not address how to approach settings where the source and target task have differing label dimension, which is the problem considered in our work and addressed by our projection approach. We also note that the authors explicitly distinguish between the setting considered in their work and the transfer learning setting, which we consider in our work, and indicate that new methods or extensions would be needed for the transfer learning setting (see the introduction and discussion of their paper).

[6] Domain Generalization via Invariant Feature Representation, ICML 2013 (reference 7 in the list below)

This work presents Domain-Invariant Component Analysis (DICA) and its unsupervised counterpart (UDICA), which are kernel-based optimization algorithms designed for domain generalization. DICA learns an invariant transformation by finding a transformation on the kernel matrix that minimizes the distributional variance (see Eq. (3) in [6]), which is a measure of dissimilarity across domains. DICA can be viewed as a generalized and enhanced version of TCA from reference [4], as evidenced by their learning of a projection in feature space (see Eq. (5) in [6]), and the connection between distributional variance and distance between empirical means. Consequently, the computational limitations described above for [4] also apply for this work, i.e., we cannot materialize the full kernel matrix on our hardware and the solver required for this method is an order of magnitude slower than using our methodology with EigenPro. In addition, we note that this method, like that of [5], can only handle problems in which the source and target tasks have the same label dimension, which is in contrast to our projection method.

[7] Multiple Kernel Learning Algorithms, JMLR 2011 (reference 8 in the list below)

This work presents a taxonomy and review of multiple kernel learning (MKL) algorithms, which combine multiple kernels to improve performance over using a single kernel. Examples of MKL algorithms include learning linear or nonlinear combinations of kernels and aligning kernels with the ideal kernel matrix given by outer products of labels. This work does not directly discuss how to use MKL for transfer learning applications and more broadly focuses on using MKL to improve performance of kernel methods in general. The main issue with using these methods to improve kernels in our work is again of computational nature, since these methods require materializing the full kernel matrices.

To clarify the contributions of our work and how it fits into the broader research context, we added the following text in the **Introduction** section summarizing prior works and the conceptual differences to our proposed method: “Prior works on transfer learning with kernels focus on applications in which the source and target tasks have the same label sets 6, 7, 9, 12... There are also various works on using kernels for multi-task learning problems 2, 3, 13, which, in the context of transfer learning, assume that source and target data are available at the time of training the source model. These methods can be computationally expensive since they involve computing matrix-valued kernels, where the number of rows/columns is equal to the number of labels. As a consequence, for a kernel trained on ImageNet32 14 with 1000 possible labels, a matrix-valued kernel would involve 10^6 times more compute than a classical kernel method. Prior works also develop kernel-based methods for learning a re-weighting or transformation that captures similarities across source and target data distributions 4, 5, 15. Such a transformation is typically learned by solving an optimization problem that involves materializing the full training kernel matrix, which can be computationally prohibitive (e.g., for a dataset with a million samples, this would require more than 3.5 terabytes of memory).”

2. Second, there is no clear specific motivation for the proposed "projection" and "translation" operations. What motivated the authors to consider these two operations specifically for transfer learning with kernel methods? They are natural and somewhat straightforward operations that anyone could come up with in the first place. Without a clear motivation, the proposed methods appear quite ad-hoc.

Our main motivation for considering these operations was their analogous, successful counterparts in transfer learning with neural networks. For example, when transferring between source and target tasks with the same label dimension using neural networks, a standard approach is to tune the weights of the source model on the target task. In the particular case of infinitely wide neural networks, which correspond to kernel machines via the Neural Tangent Kernel [16], this process is equivalent to performing translation with a kernel method (as we show in Proposition 1 in Appendix D). Our method for projection is similarly motivated by the analogous procedure in transfer learning with neural networks when the source and target tasks have differing label dimension. Namely, projection corresponds to adding layers at the end of the source network and then training the weights in these new layers on the target task.

To clarify the motivation behind our proposed methods, we added the following text to the **Introduction** section in the revised manuscript: "Projection and translation are motivated by operations that are standardly used for transfer learning using neural networks. Namely, projection corresponds to adding layers at the end of a neural network trained on a source task and then training the weights in these new layers on the target task. And when approximating a neural network by its linearization around the initial parameters [16, 17], then transfer learning by tuning the weights of the source model on the target task is equivalent to performing translation."

Furthermore, as stated on page 4 that "Definition 1 is applicable to any machine learning method, [...]", it looks like the proposed "projection" and "translation" operations may not have anything specific to do with kernel methods. What I expect to see in these two operations is some sort of technical innovation specific to kernel methods. Otherwise, what kind of gains or benefits would the kernel methods bring in this context?

The gains and benefits that kernel methods bring in this context are: (1) they are conceptually simple, effective, and flexible given that they for example include infinitely wide neural networks [16]; and (2) they provide closed form solutions, which enable a theoretical analysis of transfer learning (in the revised manuscript we extended the theoretical analysis of transfer learning using translation from the linear setting to the nonlinear setting). To clarify the reasons for our focus on kernel methods, we added the following sentences to the introduction: "We note that while these two operations are general and can be applied to any predictor, we focus on using them in conjunction with kernel methods due to their conceptual simplicity, effectiveness, and flexibility, in particular given that they include infinite width neural networks [16] as a subclass. Moreover, the closed-form solutions provided by kernel methods also enable a theoretical analysis of transfer learning."

3. Lastly, while the theoretical results are insightful, they rely on the "linearity" assumption which I believe is too simplistic in this setting. The paper claims that these theoretical results explain the empirical findings well, but I have some doubts about this; see my comments/questions below.

In the meantime, we were able to extend our result for translation from linear kernels to nonlinear kernels, showing that also in this case the risk of the translated model is reduced when the norm between the true target predictor and the source predictor is small. An extension to the non-linear setting for the projection operation is difficult, since our analysis involves the composition of two estimators, which is in contrast to other analyses of generalization in over-parameterized linear models [18, 19] that all involve only one estimator. Although our results for transfer learning using the projection operation are only for the linear setting, we believe that they are an important step towards building intuition, in particular since our results are exact and non-asymptotic as compared to related analyses that only provide bounds; see e.g. [20-24].

We added the extension to the non-linear setting for the translation operation in **Appendix E** in the revised manuscript:

"In the following, we extend Theorem 2 to kernel regression with nonlinear kernels under the assumption that the true target function is linear in the Hilbert space under the given kernel feature map.

Theorem 4. Let $\mathcal{X} = \mathbb{R}^d$, $\mathcal{Y}_s = \mathcal{Y}_t = \mathbb{R}$, let \mathcal{H} be a Hilbert space, and let $\psi : \mathcal{X} \rightarrow \mathcal{H}$ induce a positive semi-definite, bounded kernel $K(x, z) = \langle \psi(x), \psi(z) \rangle$. Let $f_s(x) = \psi(x^T)\psi(X_s)\alpha_s$ denote the source model corresponding to solving kernel regression on source data $(X_s, y_s) \in \mathcal{X}^{d \times n} \times \mathcal{Y}_s^{c \times n}$ with $\psi(X_s) =$

$[\psi(x_s^{(1)})|\psi(x_s^{(2)})|\dots|\psi(x_s^{(n)})]$. Assume the oracle target function, f^* , has the form $f^*(x) = \langle w^*, \psi(x) \rangle_{\mathcal{H}}$. Let \hat{f} denote the solution to kernel regression on the target data and let \hat{f}_t denote the translated solution. Then for target data $(X_t, y_t) \in \mathbb{R}^{d \times n} \times \mathbb{R}^{c \times n}$, it holds that

$$\begin{aligned}\mathcal{R}(\hat{f}) &\leq \mathbb{E}_{X_t} [\|(I - P)(w^*)\|_{\mathcal{H}}^2] \lambda, \\ \mathcal{R}(\hat{f}_t) &\leq \mathbb{E}_{X_t} [\|(I - P)(w^* - \psi(X_s^T)\alpha_s)\|_{\mathcal{H}}^2] \lambda,\end{aligned}$$

where $P = \psi(X_t)(\psi(X_t)^T\psi(X_t))^{-1}\psi(X_t)^T$ is a self-adjoint, compact operator and $\lambda = \mathbb{E}_x[K(x, x)]$.

Proof. We will prove the statement for the translated predictor since the corresponding statement for the base predictor follows immediately from $f_s = 0$. The translated predictor \hat{f}_t can be written as

$$\hat{f}_t(x) = K(x, X_t)K_t^{-1}y_t^T - K(x, X_t)K_t^{-1}f_s(X_t) + f_s(x) = \psi(x^T)\beta,$$

for $\beta = \psi(X_t)K_t^{-1}y_t^T - \psi(X_t)K_t^{-1}f_s(X_t) + \psi(X_s)\alpha_s \in \mathcal{H}$. Thus, the risk of this predictor is given by

$$\begin{aligned}\mathcal{R}(\hat{f}_t) &= \mathbb{E}_{x, X_t} [(f^*(x) - \hat{f}_t(x))^2] \\ &= \mathbb{E}_{x, X_t} [(\langle \psi(x), w^* \rangle - \langle \psi(x), \beta \rangle)^2] \\ &= \mathbb{E}_{x, X_t} [(\langle \psi(x), w^* - \beta \rangle)^2] \\ &\leq \mathbb{E}_{x, X_t} [\|w^* - \beta\|^2 \|\psi(x)\|^2] \quad (\text{by Cauchy-Schwarz}) \\ &= \mathbb{E}_{x, X_t} [\|w^* - \beta\|^2 K(x, x)] \\ &= \lambda \mathbb{E}_{X_t} [\|w^* - \beta\|^2].\end{aligned}$$

Now it remains to be shown that $(w^* - \beta) = (I - P)(w^* - \psi(X_s)^T\alpha_s)$. We simplify $w^* - \beta$ directly as follows:

$$\begin{aligned}w^* - \beta &= w^* - \psi(X_t)K_t^{-1}y_t^T + \psi(X_t)K_t^{-1}f_s(X_t) - \psi(X_s)\alpha_s \\ &= w^* - \psi(X_t)K_t^{-1}\psi(X_t^T)w^* + \psi(X_t)K_t^{-1}\psi(X_t^T)\psi(X_s)\alpha_s - \psi(X_s)\alpha_s \\ &= (I - P)w^* - (I - P)\psi(X_s)\alpha_s \\ &= (I - P)(w^* - \psi(X_s)\alpha_s),\end{aligned}$$

which concludes the proof. \square

Theorem 4 provides a natural extension to Theorem 2 since it shows that the risk of the translated predictor is reduced when the parameters of the source function, $\psi(X_s)\alpha_s$, and those of the true target function, w^* , are close in norm."

To sum up, the paper proposes two simple operations for transfer learning with kernel methods that show promising empirical results. However, the paper lacks well-articulated motivation and principles behind the proposed operations. It also remains unclear what we gain by focusing on kernel methods. The theoretical results are insightful but rely on an unrealistic linearity assumption.

In response to the reviewer's comments, we extended the introduction with a motivation for the two operations, provided a rationale for focusing on kernel methods, and an extension of our results to nonlinear kernels in the context of translation. In addition, as we described above, prior multiple kernel learning methods cannot scale to the datasets considered in our work. To further demonstrate the practical value of our methodology, we added an empirical comparison of our translation operation and prior kernel methods on a small binary classification task with 10,000 training examples to which prior methods can be applied. These additional experiments are discussed below (see Fig. II). We hope that these changes address all the reviewer's concerns.

=== Some questions and suggestions:

- First paragraph of Introduction: "However, developing an algorithm for transfer learning with kernel methods for general source and target tasks has been an open problem." In what sense is this an open problem? For example, transfer learning with kernel methods has been studied extensively in the domain adaptation (DA) literature. Also, since for kernel methods, we can write $f(x)$ as $f(x) = w.T\phi(x) =$

$w.Tk(x, \cdot)$, one can consider the following baseline approach: First, we train both w and kernel $k(\cdot, \cdot)$ on the source data set using "kernel learning" procedure. Then, we re-learn only the predictor w on the target data set while keeping the learned kernel fixed. This simple baseline can indeed be viewed as a transfer learning with kernels and has been adopted in the past. How does it compare with the proposed method?

Following the reviewer’s comment, we compared the suggested transfer learning method to our transfer learning approach on a variety of tasks, thereby demonstrating the effectiveness of our method. In particular, in the revised manuscript, we added a comparison between our transfer learning approach and prior multiple kernel learning algorithms [25, 28] on a small binary classification task with 10,000 examples. We used the library provided by [29] to run the prior methods. As discussed in the literature review above, these methods take orders of magnitude longer than our translation approach; for example, PWMK takes 1543 seconds (25 minutes) just to construct the kernel matrix for this simple task. Moreover, computation time scales linearly with the number of classes since these methods use one versus all approaches for multi-class classification. The results are shown in Fig. 4 below (which we added to the revised manuscript as SI Fig. 7). To describe this figure, we added the following sentence in the **Results** section: "In SI Fig 7, we additionally demonstrate the effectiveness of our translation methodology over prior transfer learning methods using Multiple Kernel Learning (see Methods for further details)."

In addition, we added the following text to the **Methods** section describing the experiments in more detail:

“Comparison with multiple kernel learning approaches. In SI Fig. 7, we compare our translation approach to the approach of directly using multiple kernel learning algorithms such as Centered Kernel Alignment (CKA) [25], EasyMKL [26], FHeuristic [27], and Proportionally Weighted Multiple Kernels (PWMK) [28] to learn a kernel on the source task and train the target model using the learned kernel. Due to the computational limitations of these prior methods, we only consider the subtask of classifying cars and deer in CIFAR10 (source) and transferring to the 19 corruptions in CIFAR10-C (target). The source task contains 10,000 training samples and 2000 test samples, while the 19 target tasks each contain 1000 training samples and 1000 test samples. The multiple kernel learning algorithms learn combinations of a Laplace kernel with bandwidth 10, a Gaussian kernel of the form $K_G(x, z) = \exp(-\gamma\|x - z\|^2)$ with $\gamma = 0.001$, and the linear kernel $K_L(x, z) = x^T z$. We choose the bandwidth for the Laplace kernel from [1] and the value of γ for the Gaussian kernel so that the entries of the Gaussian kernel are on the same order of magnitude as the Laplace kernel for this task. For each kernel learning algorithm, we train a source model and then compare the following three models on the target task: (1) the baseline model in which we use the kernel learning algorithm directly on the target task; (2) the transfer learned kernel model in which we use the weights from the source task to combine the kernels on the target task; and (3) translating the learned kernel on the source task using our translation methodology. As shown in SI Fig. 7, the transfer learned kernel outperforms the baseline kernel for almost all multiple kernel learning algorithms (except FHeuristic), and it is outperformed by our translation methodology in all cases.

Remark. We presented a comparison on this simple binary classification task for the following computational reasons. First, we considered binary classification, since prior multiple kernel learning approaches implemented in MKLPy scale poorly to multi-class problems. While there is no computational price to be paid for our method for multi-class classification, prior methods build one kernel per class and thus require 10 times more compute and memory when using all 10 classes in CIFAR10. Secondly, we compared with only translation and not projection since prior multiple kernel learning methods scale poorly to the ImageNet32 dataset used in our projection experiments. Namely, multiple kernel learning methods require materializing the kernel matrix, which for ImageNet32 would take up more than 3.5 terabytes of memory as compared to 128 gigabytes for our method.”

- Caption of Figure 1: Typo, "cell linee" → "cell lines"

We thank the reviewer for noticing this, and we updated the text accordingly.

- Page 2: "[...] translate to adding a new layer to the end of a neural network." This is unclear. Can the authors elaborate?

This sentence explains what the projection methodology corresponds to in the context of neural networks; namely, training a source network and then transfer learning by appending new layers at the end of the network and training these on the target task. More precisely, given a source network $f : \mathbb{R}^d \rightarrow \mathbb{R}^{c_s}$ of the form $f_s(x) = W_L \phi(W_{L-1} \phi(\dots W_2 \phi(W_1 x) \dots))$ with parameters $\{W_i\}_{i=1}^L$ and

(a)			
CKA Source Model Accuracy: 93.55%			
	CKA Baseline	CKA Transfer	CKA Translation
Average Acc. over 19 Corruptions	84.92%	87.05%	92.01%

(b)			
EasyMKL Source Model Accuracy: 91.95%			
	EasyMKL Baseline	EasyMKL Transfer	EasyMKL Translation
Average Acc. over 19 Corruptions	65.29%	76.79%	87.55%

(c)			
FHeuristic Source Model Accuracy: 91.65%			
	FHeuristic Baseline	FHeuristic Transfer	FHeuristic Translation
Average Acc. over 19 Corruptions	75.36%	73.81%	85.19%

(d)			
PWMK Source Model Accuracy: 91.50%			
	PWMK Baseline	PWMK Transfer	PWMK Translation
Average Acc. over 19 Corruptions	69.83%	70.24%	81.53%

Figure 1: (which corresponds to SI Fig. 7 in the revised manuscript) Comparison between (1) the baseline model in which we use the kernel learning algorithm directly on the target task; (2) the transfer learned kernel model in which we use the weights from the source task to combine the kernels on the target task; and (3) translating the learned kernel on the source task using our translation methodology. We train a source model on the binary classification task of cars versus deer in CIFAR10 (10000 training examples) and compare average test target accuracy when transferring to the classification task of cars versus deer under 19 corruptions from CIFAR10-C (1000 training, 1000 test examples). We use multiple kernel learning to combine Laplace kernel, Gaussian kernel and linear kernel and compare with four multiple kernel learning algorithms implemented in [29]: (a) Centered Kernel Alignment (CKA), (b) EasyMKL, (c) FHeuristic, and (d) Proportionally Weighted Multiple Kernels (PWMK).

element-wise nonlinearity ϕ , projection to the target task can be viewed as training the parameters $\{U_i\}_{i=1}^L$ in a model $g: \mathbb{R}^{c_s} \rightarrow \mathbb{R}^{c_t}$ of the form $g(z) = U_L \phi(U_{L-1} \phi(\dots U_2 \phi(U_1 z) \dots))$ where we consider $z = f_s(x)$. We edited the sentence in the **Introduction** as follows to clarify: “Projection and translation are motivated by operations that are standardly used for transfer learning using neural networks. Namely, projection corresponds to adding layers at the end of a neural network trained on a source task and then training the weights in these new layers on the target task...”

- In the introduction, it is unclear what motivated the authors to consider specifically projection and translation for transfer learning. Can the authors elaborate at least at an intuition level?

Our main intuition around developing these algorithms was considering their analogous operations in transfer learning with neural networks, which have been successful in practice. We added the following text to the introduction to clarify this motivation further.

Added to Introduction: “Projection and translation are motivated by operations that are standardly used for transfer learning using neural networks. Namely, projection corresponds to adding layers at the end of a neural network trained on a source task and then training the weights in these new layers on the target task. And when approximating a neural network by its linearization around the initial parameters [16, 17], transfer learning by tuning the weights of the source model on the target task is equivalent to performing translation.”

- Definition 1: What is the motivation and/or principle behind this? What makes this operation specific to kernel methods? What kind of benefits does it bring by focusing on kernel methods?

In addition to our motivation for defining the projection operator based on commonly used neural network algorithms for transfer learning, the intuition is that the output of the source model on target examples can already capture some useful information about the target data, which can then be further enhanced by training another predictor from the output of the source model to the target output. As an intuitive example, consider the schematic shown in Fig. 1a and b. While data from SVHN is not present in ImageNet32, if a source model predicts that all target images with the digit 1 are classified as flagpoles, then we can leverage projection to learn a map from the label for flagpoles to the label

for digit 1. As also described above, our reasons for focusing on kernel methods are: (1) these models are conceptually and computationally simple enabling theoretical analysis and scaling to problems that would otherwise be difficult to achieve; (2) recent advances connecting infinitely wide neural networks to a kernel model (Neural Tangent Kernel) have shown that these models are very flexibility and can be used to tackle various unstructured data tasks such as image classification; and (3) these models produce state-of-the-art results on a wide variety of tasks including drug screening [30, 31].

- In the first set of experiments on ImageNet-32, did the authors already mention somewhere that the training/validation data in ImageNet-32 do not leak into the target datasets, e.g., CIFAR10, Oxford 102 Flower, etc. Otherwise, the reported results would be problematic.

Yes, we mentioned this in the following line “It is remarkable that this performance increase is observed even for transferring to Oxford 102 Flowers or DTD, datasets that have little to no overlap with images in ImageNet32.” In particular, we note that there is no overlap between ImageNet32 and DTD or SVHN and there are only two classes with flower images in ImageNet32, yellow lady’s slipper and daisy, of which the former is not in Oxford 102 Flower and the latter is split into two sub-classes in Oxford 102 Flower.

- Figure 2b (ImageNet-32 to Flower-102): Why does the gap in accuracy between transferred kernels baselines appear to widen as the number of target examples increases? This is counterintuitive as this gap should instead reduce as we observe more target examples.

We agree that in the limit, as the number of target samples goes to infinity, this gap will reduce and the two curves will meet. However, this gap can increase in a data scarce regime with insufficient samples in the target dataset to achieve good performance with a model trained only on the target task. This is the case in the Oxford 102 Flowers dataset, where there are only 1020 training samples across 102 classes.

- Definition 2: Again, what makes this operation special for kernel methods? What kind of benefits does it bring by focusing on kernel methods?

As also described above, in addition to the simplicity and effectiveness of kernel methods, we focus on kernel methods in the context of translation, since this approach is equivalent to one of the standard transfer learning procedures for neural networks when using their linearization. Given that neural networks have been very effective for transfer learning and the NTK has been shown to outperform neural networks on various tasks, focusing on translation with kernels offers the opportunity to develop new, computationally simple state-of-the-art methods for transfer learning.

- Page 6: ”We note that training translated predictors is also a new form of boosting.” Can the authors elaborate?

Boosting involves training a predictor on the error incurred by a weak predictor. More precisely, given a dataset $(X, y) \in \mathbb{R}^{d \times n} \times \mathbb{R}^{c \times n}$ and a non-interpolating predictor f_1 trained on (X, y) , we can perform boosting by training a model, f_2 on $(X, y - f_1(X))$ and then utilizing the joint predictor $f_1 + f_2$. By definition, we cannot perform boosting with interpolating kernel methods on a fixed dataset since $f_1(X) = y$ for any interpolating model. In the context of transfer learning, given a source predictor f_s and target data (X_t, y_t) , we can perform boosting by training a target model f_c on the dataset $(X_t, y_t - f_s(X))$ and then utilizing the predictor $f_s + f_c$. This approach is equivalent to translation as described in Definition 2.

- Page 7: ”Overall, our results showcase that translation is effective at adapting kernel methods to distribution shift in image classification.” The conclusion is specific to kernel methods, while it remains unclear what makes Definition 2 special for kernel methods.

We chose to keep the conclusion specific to kernel methods, since our empirical results all use kernel methods. We would like to clarify that it is not that Definition 2 is special for kernel methods, but rather that Definition 2 enables transfer learning with kernel methods (a particularly effective class of models) in a manner analogous to current effective transfer learning approaches for neural networks. Moreover, our newly added results in SI Fig. 7 (Fig. 4 in this response) demonstrate that our proposed approach motivated by neural network methodology is more effective than prior methods for domain adaption with kernels based on multiple kernel learning.

- Page 9: ”[...] the concatenation above, we get the translated predictor.” How? I tried, but I cannot

see how we can get the translated predictor because \hat{f}_s disappears from the equation.

We thank the reviewer for this question. The text should say that f_{pt} reduces to translation if the projection function is additive in its arguments. We corrected the sentence in the **Results** section as follows: “Note that if we omit x, X_t in the concatenation above, we get the projected predictor, and if f is additive in its arguments, i.e., if $f([\hat{f}_s(x) | x]) = \hat{f}_s(x) + x$, we get the translated predictor.”

- Page 9: “[...] the performance gap between the transferred NTK and the baseline NTK grows as more target examples are collected.” Can the authors elaborate more on this behavior? This is related to one of my previous questions.

The scaling law implies that the performance of the transfer learned model will grow faster than that of the target. As a consequence, it is expected that the gap initially grows larger as more samples are collected. Yet, performance is limited by a max R^2 of 1. Hence, assuming we are using a statistically consistent estimator, after a certain number of target examples are collected, these two curves must meet, and thus the gap must reduce. In the revised manuscript, we aimed to clarify this point by updating the corresponding sentence in the **Results** section as follows: “We note that the transferred NTKs have better scaling coefficients than the baseline models, thereby implying that the performance gap between the transferred NTK and the baseline NTK grows as more target examples are collected until the performance of the transferred NTK saturates at its maximum possible value.”

- Page 9: Typo, “[...] for other transfer learning tasks [...]” → “[...] for other transfer learning tasks [...]”

We thank the reviewer for noticing this, and we updated the text accordingly.

- Theoretical analysis: Why did the authors make the linearity assumption? It is highly unrealistic in the context of this work. Do the analyses make sense for classification problems or the focus is on regression problems?

We made this linear assumption for the following two reasons: (1) it is important to develop the relevant theory for the linear setting first in order to gain intuition for the nonlinear setting; (2) the linear setting allows us to leverage recent results in random matrix theory to calculate the risk *exactly*, as opposed to just obtaining *asymptotic bounds* on the risk. We have now provided an extension to the nonlinear setting for translation assuming that the true solution is linear after the fixed feature map of the kernel (we added this result as Theorem 4 in Appendix E in the revised manuscript). We would also like to point out that to the best of our knowledge our theory in the linear setting is novel even in light of recent results on understanding the risk of over-parameterized models in linear regression [18, 19, 32]. While the analyses in our work are primarily for the regression setting, they can be used to gain intuition also for the classification setting given that risk is a commonly used surrogate loss for classification accuracy [33].

- Theorem 1: How does this result explain the “logarithmic” scaling law observed in the empirical results? Where does the logarithmic dependency appear in the theoretical result?

Our linear results do not explain the logarithmic trend in the empirical results, which we envision will require precise bounds in the nonlinear setting. Nevertheless, our theoretical results serve to provide key intuition and capture unexpected dependencies on the number of source labels, target labels, source samples, and target samples, which were both observed empirically in prior work [34] and in our experiments.

- Page 10, Remark: “[...] corroborate surprising dependencies [...]” Why is this surprising?

An example of a surprising dependency which we refer to is that increasing the number of source classes while keeping the number of examples per class fixed can lead to worse transfer with projection. This is a priori unexpected given generalization bounds for multi-task learning problems [20, 22] since these works argue that in the multi-task setting, increasing the number of tasks decreases the overall risk. However, our non-asymptotic analysis demonstrates that this decrease in risk only holds as the number of classes *and* the number of examples per class increases. To clarify this, we added the following sentences to the remark: “However, note that such behavior is a priori unexpected given generalization bounds for multi-task learning problems [20, 22], which show that increasing the number of tasks decreases the overall risk. Our non-asymptotic analysis demonstrates that such decrease in risk only holds as the number of classes *and* the number of examples per class increase.”

- Theorem 2: Typo, "the the" → "the"

We thank the reviewer for noticing this, and we updated the text accordingly.

- Page 11: "The proof is given in SI Section E" The period is missing at the end of the sentence.

We thank the reviewer for noticing this, and we updated the text accordingly.

- Page 11, Discussion: Typo, "empirtically" → "empirically", "drastially" → "drastically"

We thank the reviewer for noticing this, and we updated the text accordingly.

- Page 12: "[...] we used a bandwidth of $L=10$." How did the authors choose this hyperparameter? Did the authors perform model selection?

We used this hyperparameter since it was used in prior works [35] (selected via cross-validation) for the analysis of the CIFAR10 and SVHN datasets. We added the following sentence in the revised **Methods** section to clarify this: "For our experiments, we used a bandwidth of $L = 10$ as in [35], selected through cross-validation."

- Page 13: "[...] the CNN was trained using Adam with a learning rate of 10^{-4} ." How did the authors choose this hyperparameter?

We used this hyperparameter since it has been used in several previous works [36, 38] and did not lead to divergence in the training loss. We added the following sentence in the revised **Methods** section to clarify this: "Our choice of learning rate is based on its effectiveness in prior works [36, 37]."

- Page 13: Would the proposed methods still applicable if we do not have access to these aggregated class indices in practice?

When we do not have access to the aggregated class indices, we would use projection, since in this case the source and target data have different label dimensions. We used aggregated classes for this example involving ImageNet32 and CIFAR10, since the translation operation requires the source and target classes to have the same dimension, and we wanted to provide an additional example demonstrating the effectiveness of translation.

- Caption of Figure 9: Typo, "[...] examples per class increases.." → "[...] examples per class increases."

We thank the reviewer for noticing this, and we updated the text accordingly.

Reviewer #3 response:

The paper is on an important topic - transfer learning - and the authors have done very good work both theoretically and experimentally. The experimental results shown are also strong. They consider two types of transfer learning - with the source and target tasks having the same or different label sets.

I have two major concerns, hence recommending a major (risky) revision.

We thank the reviewer for their thoughtful and detailed feedback on our paper and for the positive comments on the strength and quality of our theoretical and experimental results. We hope our point-by-point response below together with the addition of an experimental comparison with prior works on multiple kernel learning methods in Fig. 1 above (which we added as SI Fig. 7 in the revised manuscript) address the reviewer's concerns.

First, the authors unfortunately partly “reinvent the wheel”. While they do not refer to earlier work, some of the ideas they propose (e.g. the additive approach for the case of source and target tasks having the same labels - definition 2, page 6) is very similar to work proposed since the early 2000s - see these articles and references therein (also to earlier work on multi-task and transfer learning):

[1] T. Evgeniou, M. Pontil. Regularized multi-task learning. Proceedings of the tenth ACM SIGKDD international conference on Knowledge, 2004.

[2] T. Evgeniou, C.A. Micchelli, M. Pontil. Learning multiple tasks with kernel methods. Journal of Machine Learning Research, 2005

[3] A. Argyriou, T. Evgeniou, M. Pontil. Convex multi-task feature learning. Machine Learning 73 (3), 243-272, 2008.

[4] Tommasi, T., Orabona, F., and Caputo, B. Safety in Numbers: Learning Categories from Few Examples with Multi Model Knowledge Transfer. CVPR 2010.

The authors should compare with - and build upon - these series of works. It is not clear how major are the proposed innovations relative to that - and other - work, hence the recommendation is for a major revision (not a rejection - hoping the authors will provide significant advances in this important topic).

We thank the reviewer for pointing out these prior works. We note that the computational complexity of these prior methods prevents running these methods on the datasets presented in our work. In particular, the works [1, 2] require computing matrix-valued kernels, which for the case of ImageNet32, would require 10^6 more computations per iteration of EigenPro than our method. Such computation would take well over a year even for a large cluster of GPUs as opposed to the few days of computation on a single server with our method. The works [3, 4] involve explicitly computing the square root or inverse of a matrix of size at least $m \times m$ where m is the number of training examples for a task. For ImageNet32 with $m > 10^6$, this matrix is infeasible to even store on our setup since it would require over 3.5 terabytes of memory (as compared to 128 gigabytes for our method). In the following, we briefly summarize each of the suggested related works and describe their computational limitations as well as conceptual differences to our work in greater detail.

[1] T. Evgeniou, M. Pontil. Regularized multi-task learning. Proceedings of the tenth ACM SIGKDD international conference on Knowledge, 2004. (reference 13 in the list below)

This work presents an approach for multi-task learning based on the assumption that predictors on tasks can be decomposed into a shared component and predictor specific component. The work then develops kernel methods to learn these components. As stated by the authors, the runtime of the proposed method is $O(T^3m^3)$, where T is the number of tasks and m is the number of examples per task. In the case of transfer learning from ImageNet32, the number of tasks T is over 1000 (since there is one task per source and target class) and the number of samples is over 10^6 . In fact, the authors state that “Optimizing the running time of the multi-task learning method we propose is therefore an important practical issue that we leave as an open problem.” Even if one were to use EigenPro to solve kernel regression instead of the SVM proposed in this work, it would require 10^6 times more computation time than our proposed method, since this method would need to compute a kernel entry for every combination of label and example. Computation with this method on the ImageNet32 task would take well over a year as compared to the few days of computation required with our method.

[2] T. Evgeniou, C.A. Micchelli, M. Pontil. Learning multiple tasks with kernel methods. *Journal of Machine Learning Research*, 2005 (reference [3] in the list below)

This work investigates learning multiple related tasks simultaneously using kernel methods and regularization by extending single-task kernel methods to multi-task learning. The authors develop a family of multi-task kernel functions to model task relationships and demonstrate improved performance over standard single-task learning in cases with numerous related tasks and limited data per task. Since this work employs matrix-valued kernels for multi-task learning, the computational limitations of [1] apply also to this work. In particular, the experiments that we performed on ImageNet32 would take over a year using this method when running on the same hardware as used in our work.

[3] A. Argyriou, T. Evgeniou, M. Pontil. Convex multi-task feature learning. *Machine Learning* 73 (3), 243-272, 2008. (reference [15] in the list below)

This work presents alternating minimization algorithms for learning a shared feature map across many source tasks. Empirical results show that the method can learn common features across tasks, improving performance compared to single-task learning. This work involves computing matrix square roots of matrices of size $\delta \times \delta$, where δ is the dimension of the orthogonal basis of the training data after the feature map. For the multi-class classification tasks considered in our work, $\delta = m$, where m is the number of training samples. As noted by the authors, computing an explicit matrix square root requires $O(\delta^3)$ runtime and $O(\delta^2)$ memory. For the case of ImageNet32 considered in our work, this would require over 3.5 terabytes of memory, making the corresponding matrix square root computation infeasible on our hardware. In addition to these computational limitations, the authors also state that in their experiment, “a certain number of tasks (greater than 10)” was required for a boost over independent learning. This is not the case with our methodology; the experiments shown in Fig. [1] show a significant boost even when the source and target model have just one task.

[4] Tommasi, T., Orabona, F., and Caputo, B. Safety in Numbers: Learning Categories from Few Examples with Multi Model Knowledge Transfer. *CVPR 2010*. (reference [12] in the list below)

This work proposes a kernel method for transfer learning between image data by constraining the target model to be similar to one of the pre-trained source models through an explicit regularization term. The authors state that the runtime of their algorithm is $O(\ell^3 + k\ell^2)$ where ℓ is the number of datapoints and k is the number of binary classification source tasks. This runtime is dominated by the required explicit inversion of the regularized training kernel matrix. Our method does not require such inversion and the runtime is only $O(t\ell^2)$, where t is the number of rounds of EigenPro ($t \leq 10$ for our projection experiments). In addition, this prior work considers settings in which the source and target datasets have the same label dimension, whereas our projection method can handle tasks with differing label dimension.

To clarify the contributions of our work and how it fits into the broader research context, we added the following text in the **Introduction** section summarizing prior works and the conceptual differences to our proposed method: “Prior works on transfer learning with kernels focus on applications in which the source and target tasks have the same label sets [6, 7, 9, 12]... There are also various works on using kernels for multi-task learning problems [2, 3, 13], which, in the context of transfer learning, assume that source and target data are available at the time of training the source model. These methods can be computationally expensive since they involve computing matrix-valued kernels, where the number of rows/columns is equal to the number of labels. As a consequence, for a kernel trained on ImageNet32 [14] with 1000 possible labels, a matrix-valued kernel would involve 10^6 times more compute than a classical kernel method. Prior works also develop kernel-based methods for learning a re-weighting or transformation that captures similarities across source and target data distributions [4, 5, 15]. Such a transformation is typically learned by solving an optimization problem that involves materializing the full training kernel matrix, which can be computationally prohibitive (e.g., for a dataset with a million samples, this would require more than 3.5 terabytes of memory).”

With respect to experimental comparisons of our proposed method to prior kernel methods: Given the severe computational limitations of the above described prior methods, we could only compare to related prior methods on multiple kernel learning on a small dataset. Fig. [1] above (which is SI Fig. 7 in the revised manuscript) shows a comparison between our transfer learning approach and prior multiple kernel learning algorithms [25-28] on a small binary classification task with 10,000 examples. We used

the library provided by [29] to run these methods. These methods take orders of magnitude longer than our translation approach; for example, PWMK takes 1543 seconds (25 minutes) just to construct the kernel matrix for this simple task. Moreover, computation time scales linearly with the number of classes since these methods use one versus all approaches for multi-class classification. We added the following sentence in the **Results** section to describe these experiments: “In SI Fig 7, we additionally demonstrate the effectiveness of our translation methodology over prior transfer learning methods using multiple kernel learning (see Methods for further details).

In addition, we added the following text to the **Methods** section describing the experiments in more detail:

“Comparison with multiple kernel learning approaches. In SI Fig. 7, we compare our translation approach to the approach of directly using multiple kernel learning algorithms such as Centered Kernel Alignment (CKA) [25], EasyMKL [26], FHeuristic [27], and Proportionally Weighted Multiple Kernels (PWMK) [28] to learn a kernel on the source task and train the target model using the learned kernel. Due to the computational limitations of these prior methods, we only consider the subtask of classifying cars and deer in CIFAR10 (source) and transferring to the 19 corruptions in CIFAR10-C (target). The source task contains 10,000 training samples and 2000 test samples, while the 19 target tasks each contain 1000 training samples and 1000 test samples. The multiple kernel learning algorithms learn combinations of a Laplace kernel with bandwidth 10, a Gaussian kernel of the form $K_G(x, z) = \exp(-\gamma\|x - z\|^2)$ with $\gamma = 0.001$, and the linear kernel $K_L(x, z) = x^T z$. We choose the bandwidth for the Laplace kernel from [1] and the value of γ for the Gaussian kernel so that the entries of the Gaussian kernel are on the same order of magnitude as the Laplace kernel for this task. For each kernel learning algorithm, we train a source model and then compare the following three models on the target task: (1) the baseline model in which we use the kernel learning algorithm directly on the target task; (2) the transfer learned kernel model in which we use the weights from the source task to combine the kernels on the target task; and (3) translating the learned kernel on the source task using our translation methodology. As shown in SI Fig. 7, the transfer learned kernel outperforms the baseline kernel for almost all multiple kernel learning algorithms (except FHeuristic), and it is outperformed by our translation methodology in all cases.

Remark. We presented a comparison on this simple binary classification task for the following computational reasons. First, we considered binary classification, since prior multiple kernel learning approaches implemented in MKLPy scale poorly to multi-class problems. While there is no computational price to be paid for our method for multi-class classification, prior methods build one kernel per class and thus require 10 times more compute and memory when using all 10 classes in CIFAR10. Secondly, we compared with only translation and not projection since prior multiple kernel learning methods scale poorly to the ImageNet32 dataset used in our projection experiments. Namely, multiple kernel learning methods require materializing the kernel matrix, which for ImageNet32 would take up more than 3.5 terabytes of memory as compared to 128 gigabytes for our method.”

Second, note that there are also theoretical results that the authors can compare with/build upon, such as:

- [5] J Baxter A model of inductive bias learning J. Artif. Intell. Res. (JAIR) 12, 149-198, 2000.
- [6] Ando, Rie Kubota, Tong Zhang, and Peter Bartlett. A framework for learning predictive structures from multiple tasks and unlabeled data.” Journal of Machine Learning Research 6, no. 11 (2005).
- [7] Maurer A, Pontil M, Romera-Paredes B. The benefit of multitask representation learning. Journal of Machine Learning Research. 2016;17(81):1-32.
- [8] I Kuzborskij, F Orabona Fast Rates by Transferring from Auxiliary Hypotheses Machine Learning, 2017.
- [9] Denevi, C Ciliberto, D Stamos, M Pontil Learning to learn around a common mean Advances in Neural Information Processing Systems 31, 2018.
- [10] M Khodak, MFF Balcan, AS Talwalkar Adaptive gradient-based meta-learning methods Advances in Neural Information Processing Systems 32, 2019.

We thank the reviewer for pointing out these related works and their theoretical contributions. Given that these works consider problems that significantly differ from the transfer learning setting considered in our work (namely multi-task learning in [5, 6, 7, 8] and learning to learn in [9, 10]), we focus solely on their theoretical analyses. In the following, we provide a detailed description of the theoretical analyses

performed in each of the proposed prior works and highlight the differences to our work. In particular, we note that these prior works only provide *asymptotic generalization bounds*, which, unlike our exact non-asymptotic results, cannot be used to obtain a formula for the scaling law of a predictor.

[5] J Baxter. A model of inductive bias learning. J. Artif. Intell. Res. (JAIR) 12, 149-198, 2000. (reference 20 in the list below)

This work introduces a formal model of inductive bias learning in environments with multiple related tasks, aiming to learn a hypothesis space suitable for the whole environment. The work utilizes VC-dimension based arguments to identify the number of samples and tasks needed before any element from a given hypothesis class results in low risk. We note that the VC-dimension is infinite for interpolating kernel methods such as the Gaussian kernel, and thus the bounds derived in this paper may be vacuous for the applications considered in our work. Moreover, while the bounds derived in this paper require asymptotically many samples for generalization (even for linear models), we provide non-asymptotic results in the linear case, which hold even in the over-parameterized setting with more features than samples and give rise to explicit scaling laws.

[6] Ando, Rie Kubota, Tong Zhang, and Peter Bartlett. A framework for learning predictive structures from multiple tasks and unlabeled data.” Journal of Machine Learning Research 6, no. 11 (2005). (reference 21 in the list below)

This work presents an analysis and an algorithm for structural learning, which involves identifying common structure useful for improving average performance across multiple tasks. Their theoretical results are based on generalization bounds for a hypothesis space of linear functions with shared component. In particular, the authors express the true risk across all tasks as bounded by the empirical risk plus terms that are minimized when the number of tasks goes to infinity. While such bound-based theoretical results are useful for proving consistency as the number of tasks and samples goes to infinity, they do not allow for a characterization of the underlying scaling laws. Obtaining explicit scaling laws for the linear regression setting as obtained in our work requires exact non-asymptotic results for the risk of the projected and translated predictor, which is what we concentrated on.

[7] Maurer A, Pontil M, Romera-Paredes B. The benefit of multitask representation learning. Journal of Machine Learning Research. 2016;17(81):1-32. (reference 22 in the list below)

This work presents a theoretical analysis of multitask representation learning and focuses on demonstrating the benefit of such joint learning over learning on independent tasks. The work assumes that the tasks can be decomposed into the form $f_t(h(x))$ where f_t is specific to a given task and h is a shared feature map across tasks. In this setting, the authors provide generalization bounds for a range of hypothesis spaces showing that the difference in true and empirical risk goes to zero as the number of tasks increases and the number of samples per task goes to infinity. Again, while such results are useful for understanding the asymptotic benefit of multi-task learning, the presented theoretical analysis is not sufficient for extracting an explicit scaling law.

[8] I Kuzborskij, F Orabona. Fast Rates by Transferring from Auxiliary Hypotheses. Machine Learning, 2017. (reference 23 in the list below)

This work presents a theoretical analysis of a regularized empirical risk minimization approach in which the regularization term for training the target predictor penalizes the distance between the target predictor and a linear combination of source predictors. This framework generalizes that of least-squares with biased regularization from 12. The authors provide generalization bounds for such an approach, demonstrating that the risk of the target predictor can decrease an order of magnitude faster provided that the risk of the source predictor on the target task is small. Again, since this work establishes bounds, we cannot recover precise scaling laws for the predictor from these results. In contrast, our work provides exact non-asymptotic results for the case of linear predictors. For the particular case of translation, which is a hyper-parameter free formulation (i.e., no β term) of least-squares with biased regularization, our Theorem 2 provides an explicit formula for the risk of the translated predictor in terms of the risk of the source model on the target task; namely, $\mathcal{R}(\hat{w}_t) = (1 - \frac{n_t}{d}) \mathcal{R}(\hat{w}_s)$, where $\mathcal{R}(\hat{w}_s)$ is the risk of the source model on the target dataset. Note that when $\mathcal{R}(\hat{w}_s) \propto (1 - \frac{n_t}{d}) \mathcal{R}(\hat{w}_b)$, our result provides explicit conditions under which the translated predictor can have order of magnitude decrease in risk over the baseline predictor even without requiring multiple tasks.

[9] Denevi, C Ciliberto, D Stamos, M Pontil. Learning to learn around a common mean. Advances

in Neural Information Processing Systems 31, 2018. (reference [24] in the list below)

This work presents a regression approach to learning to learn. In particular, the authors present and analyze an algorithm for learning a common mean across tasks, which is subsequently used to regularize the predictors on each task. The formulation is for linear models that are assumed to have a common mean. For these models, the authors provide generalization bounds that establish consistency of the developed method, which estimates the common mean and builds predictors for each task. Again, we note that the generalization bound techniques used do not provide explicit scaling laws, unlike our bounds, which provide exact non-asymptotic results for the linear case.

[10] M Khodak, MFF Balcan, AS Talwalkar. Adaptive gradient-based meta-learning methods. Advances in Neural Information Processing Systems 32, 2019. (reference [39] in the list below)

This work presents a framework, Average Regret-Upper-Bound Analysis (ARUBA), for analyzing learning to learn algorithms and improving upon previous gradient based meta-learning approaches. The work considers a different setting from ours, namely the authors analyze an upper bound on the regret of online learning models, i.e., they analyze the performance of the meta-learner on a series of future tasks. The authors focus on upper bounding the regret of an online learner based on the average of the upper bounds on the regret across tasks, while we directly focus on the supervised transfer learning setting in which the goal is to minimize the risk on the target task from a pre-trained source model.

To clarify the contributions of our work, We added the following sentences to the **Introduction** section, explaining that the generalization bounds from prior works are unable to provide explicit scaling laws as obtained in our work: “To do so, we obtain exact non-asymptotic formulas for the risk of the projected and translated predictors. Our non-asymptotic analysis is in contrast to a large number of prior works analyzing multi-task learning algorithms [20–23] and meta-learning algorithms [24, 39], which provide generalization bounds establishing statistical consistency of these methods but do not provide an explicit form of the risk, which is required for deriving explicit scaling laws.”

Minor: Theorem 2 in p. 10 has some typos (“the” is “then”, and I assume wb is ws? But then can we also say something about ws?)

We thank the reviewer for pointing out the typo (missing “the” in Theorem 2), which we corrected accordingly. The w_b in this theorem is not a typo since we are trying to establish how the risk of the translated predictor compares to that of the baseline predictor w_b . Indeed, Theorem 2 implies that the translated predictor has lower risk than the baseline predictor when $\|w_s - w_t\|_F$ is small, i.e., when the true source model and true target model are similar.

References

1. Ma, S. & Belkin, M. *Kernel machines that adapt to GPUs for effective large batch training in Conference on Machine Learning and Systems* (2019).
2. Micchelli, C. & Pontil, M. Kernels for Multi-task Learning. *Advances in neural information processing systems* **17** (2004).
3. Evgeniou, T., Micchelli, C. A., Pontil, M. & Shawe-Taylor, J. Learning multiple tasks with kernel methods. *Journal of machine learning research* **6** (2005).
4. Gretton, A. *et al.* Covariate shift by kernel mean matching. *Dataset shift in machine learning* **3**, 5 (2009).
5. Pan, S. J., Tsang, I. W., Kwok, J. T. & Yang, Q. Domain adaptation via transfer component analysis. *IEEE transactions on neural networks* **22**, 199–210 (2010).
6. Blanchard, G., Lee, G. & Scott, C. Generalizing from several related classification tasks to a new unlabeled sample. *Advances in neural information processing systems* **24** (2011).
7. Muandet, K., Balduzzi, D. & Schölkopf, B. *Domain generalization via invariant feature representation in International conference on machine learning* (2013), 10–18.
8. Gönen, M. & Alpaydm, E. Multiple kernel learning algorithms. *The Journal of Machine Learning Research* **12**, 2211–2268 (2011).

9. Dai, W., Yang, Q., Xue, G.-R. & Yu, Y. *Boosting for Transfer Learning* in *ACM International Conference Proceeding Series* **227** (Jan. 2007), 193–200.
10. Lin, H. & Reimherr, M. On Transfer Learning in Functional Linear Regression. *arXiv:2206.04277* (2022).
11. Obst, D. *et al.* Transfer Learning for Linear Regression: a Statistical Test of Gain. *arXiv:2102.09504* (2021).
12. Tommasi, T., Orabona, F. & Caputo, B. *Safety in numbers: Learning categories from few examples with multi model knowledge transfer* in *Computer Vision and Pattern Recognition* (2010), 3081–3088.
13. Evgeniou, T. & Pontil, M. *Regularized multi-task learning* in *Proceedings of the tenth ACM SIGKDD international conference on Knowledge discovery and data mining* (2004), 109–117.
14. Chrabaszcz, P., Loshchilov, I. & Hutter, F. A downsampled variant of imagenet as an alternative to the cifar datasets. *arXiv:1707.08819* (2017).
15. Argyriou, A., Evgeniou, T. & Pontil, M. Convex multi-task feature learning. *Machine learning* **73**, 243–272 (2008).
16. Jacot, A., Gabriel, F. & Hongler, C. *Neural Tangent Kernel: Convergence and Generalization in Neural Networks* in *Advances in Neural Information Processing Systems* (eds Bengio, S. *et al.*) (Curran Associates, Inc., 2018).
17. Liu, C., Zhu, L. & Belkin, M. *On the linearity of large non-linear models: when and why the tangent kernel is constant* in *Neural Information Processing Systems* (2020).
18. Hastie, T., Montanari, A., Rosset, S. & Tibshirani, R. J. Surprises in high-dimensional ridgeless least squares interpolation. *arXiv:1903.08560* (2019).
19. Belkin, M., Hsu, D. & Xu, J. Two models of double descent for weak features. *Society for Industrial and Applied Mathematics Journal on Mathematics of Data Science* **2**, 1167–1180 (2020).
20. Baxter, J. A model of inductive bias learning. *Journal of artificial intelligence research* **12**, 149–198 (2000).
21. Ando, R. K., Zhang, T. & Bartlett, P. A framework for learning predictive structures from multiple tasks and unlabeled data. *Journal of Machine Learning Research* **6** (2005).
22. Maurer, A., Pontil, M. & Romera-Paredes, B. The benefit of multitask representation learning. *Journal of Machine Learning Research* **17**, 1–32 (2016).
23. Kuzborskij, I. & Orabona, F. Fast rates by transferring from auxiliary hypotheses. *Machine Learning* **106**, 171–195 (2017).
24. Denevi, G., Ciliberto, C., Stamos, D. & Pontil, M. Learning to learn around a common mean. *Advances in Neural Information Processing Systems* **31** (2018).
25. Cortes, C., Mohri, M. & Rostamizadeh, A. *Two-Stage Learning Kernel Algorithms* in (Aug. 2010), 239–246.
26. Aioli, F. & Donini, M. EasyMKL: a scalable multiple kernel learning algorithm. *Neurocomputing* **169**, 215–224 (2015).
27. Qiu, S. & Lane, T. A framework for multiple kernel support vector regression and its applications to siRNA efficacy prediction. *IEEE/ACM Transactions on Computational Biology and Bioinformatics* **6**, 190–199 (2008).
28. Tanabe, H., Ho, T. B., Nguyen, C. H. & Kawasaki, S. *Simple but effective methods for combining kernels in computational biology* in *2008 IEEE International Conference on Research, Innovation and Vision for the Future in Computing and Communication Technologies* (2008), 71–78.
29. Lauriola, I. & Aioli, F. MKLpy: a python-based framework for Multiple Kernel Learning. *arXiv preprint arXiv:2007.09982* (2020).
30. Arora, S. *et al.* *Harnessing the Power of Infinitely Wide Deep Nets on Small-data Tasks* in *International Conference on Learning Representations* (2020).
31. Radhakrishnan, A., Stefanakis, G., Belkin, M. & Uhler, C. Simple, Fast, and Flexible Framework for Matrix Completion with Infinite Width Neural Networks. *arXiv:2108.00131* (2021).

32. Bartlett, P. L., Long, P. M., Lugosi, G. & Tsigler, A. Benign overfitting in linear regression. *Proceedings of the National Academy of Sciences* **117**, 30063–30070 (2020).
33. Hui, L. & Belkin, M. *Evaluation of Neural Architectures Trained with Square Loss vs Cross-Entropy in Classification Tasks* in *International Conference on Learning Representations* (2021).
34. Huh, M., Agrawal, P. & Efros, A. A. What makes ImageNet good for transfer learning? *arXiv:1608.08614* (2016).
35. Belkin, M., Ma, S. & Mandal, S. *To understand deep learning we need to understand kernel learning* in *International Conference on Machine Learning* (2018), 541–549.
36. Radhakrishnan, A., Belkin, M. & Uhler, C. Overparameterized neural networks implement associative memory. *Proceedings of the National Academy of Sciences* **117**, 27162–27170 (2020).
37. Howard, J. & Ruder, S. *Universal Language Model Fine-tuning for Text Classification* in *Association for Computational Linguistics* (Association for Computational Linguistics, 2018), 328–339.
38. Das, N., Hussain, E. & Mahanta, L. B. Automated classification of cells into multiple classes in epithelial tissue of oral squamous cell carcinoma using transfer learning and convolutional neural network. *Neural Networks* **128**, 47–60 (2020).
39. Khodak, M., Balcan, M.-F. F. & Talwalkar, A. S. Adaptive gradient-based meta-learning methods. *Advances in Neural Information Processing Systems* **32** (2019).

REVIEWERS' COMMENTS

Reviewer #2 (Remarks to the Author):

In my original review, I raised three major concerns, namely, the lack of adequate literature review in the area of transfer learning with kernel methods, unclear motivation for the proposed "projection" and "translation" operations, and the linearity assumption in the theoretical analysis.

The authors have thoroughly responded to the above concerns which I highly appreciate. First, the authors clarified the novelty and contributions of this work by providing detailed comparisons to existing methods, both in terms of writing and additional experimental results. It is now much clearer what the comparative contributions of this work are. Second, the authors clarified the motivation for the proposed "projection" and "translation" operations, attributing them to their analogous, successful counterparts in transfer learning with neural networks. Lastly, to address my concern about the linearity assumption in the theoretical analysis, the authors provided an additional theoretical result in a nonlinear setting. Although the result does not translate fully to a more general non-linear setting, I appreciate the authors' effort as obtaining the result with full generality is considered hard.

My skepticism on technical innovation that is specific to kernel methods remains, but to the extent that is specific to my own taste. Other scholars in the area might disagree with me.

The manuscript has been revised and now has better writing quality compared to the previous version.

Reviewer #3 (Remarks to the Author):

Thank you - and congratulations - for an excellent revision of the article as well as very clear responses to the first review feedback. I find the changes made to be clear and satisfactory, and therefore I fully support the publication of the article on this important topic. Thank you also for all clarifications - and for bringing in new issues and solutions in this area.

Reviewer # 2 response:

In my original review, I raised three major concerns, namely, the lack of adequate literature review in the area of transfer learning with kernel methods, unclear motivation for the proposed "projection" and "translation" operations, and the linearity assumption in the theoretical analysis.

The authors have thoroughly responded to the above concerns which I highly appreciate. First, the authors clarified the novelty and contributions of this work by providing detailed comparisons to existing methods, both in terms of writing and additional experimental results. It is now much clearer what the comparative contributions of this work are. Second, the authors clarified the motivation for the proposed "projection" and "translation" operations, attributing them to their analogous, successful counterparts in transfer learning with neural networks. Lastly, to address my concern about the linearity assumption in the theoretical analysis, the authors provided an additional theoretical result in a nonlinear setting. Although the result does not translate fully to a more general non-linear setting, I appreciate the authors' effort as obtaining the result with full generality is considered hard.

My skepticism on technical innovation that is specific to kernel methods remains, but to the extent that is specific to my own taste. Other scholars in the area might disagree with me.

The manuscript has been revised and now has better writing quality compared to the previous version.

We thank the reviewer for their positive feedback and helpful comments in improving our manuscript. We are glad to have addressed their concerns.

Reviewer # 3 response:

Thank you - and congratulations - for an excellent revision of the article as well as very clear responses to the first review feedback. I find the changes made to be clear and satisfactory, and therefore I fully support the publication of the article on this important topic. Thank you also for all clarifications - and for bringing in new issues and solutions in this area.

We thank the reviewer for their positive feedback and helpful comments in improving our manuscript. We are glad to have addressed their concerns.